# X-Transfer Attacks: Towards Super Transferable Adversarial Attacks on CLIP

Hanxun Huang [1]   Sarah Erfani [1]   Yige Li [2]   Xingjun Ma [3]   James Bailey [1]

## Abstract

As Contrastive Language-Image Pre-training (CLIP) models are increasingly adopted for diverse downstream tasks and integrated into large vision-language models (VLMs), their susceptibility to adversarial perturbations has emerged as a critical concern. In this work, we introduce **X-Transfer**, a novel attack method that exposes a universal adversarial vulnerability in CLIP. X-Transfer generates a Universal Adversarial Perturbation (UAP) capable of deceiving various CLIP encoders and downstream VLMs across different samples, tasks, and domains. We refer to this property as **super transferability**—a single perturbation achieving cross-data, cross-domain, cross-model, and cross-task adversarial transferability simultaneously. This is achieved through **surrogate scaling**, a key innovation of our approach. Unlike existing methods that rely on fixed surrogate models, which are computationally intensive to scale, X-Transfer employs an efficient surrogate scaling strategy that dynamically selects a small subset of suitable surrogates from a large search space. Extensive evaluations demonstrate that X-Transfer significantly outperforms previous state-of-the-art UAP methods, establishing a new benchmark for adversarial transferability across CLIP models. The code is publicly available in our GitHub repository.

## 1. Introduction

Contrastive Language-Image Pre-training (CLIP) is a widely adopted technique that learns aligned multi-modal representations from text-image pairs through contrastive learning (Radford et al., 2021). Pre-trained on web-scale datasets, CLIP encoders have been extensively used to enhance performance across a variety of downstream applications, particularly in large Vision-Language Models (VLMs) (Awadalla et al., 2023; Koh et al., 2023; Wang et al., 2023a; Bai et al., 2023; Karamcheti et al., 2024; Jiang et al., 2024), where they form the backbone of visual capabilities. Models such as Flamingo (Alayrac et al., 2022), LLaVA (Liu et al., 2023a), BLIP2 (Li et al., 2023a), and MiniGPT-4 (Zhu et al., 2024) integrate CLIP image encoders with Large Language Models (LLMs) (Zhang et al., 2022b; Hoffmann et al., 2022; Chiang et al., 2023). The widespread adoption of CLIP in VLMs is primarily driven by its pre-training paradigm utilising text supervision (Tong et al., 2024). While its strong generalisation capabilities solidify its role as a cornerstone for VLMs, they also make CLIP an ideal target for generating highly transferable adversarial perturbations, thereby introducing new safety risks.

Deep neural networks are widely recognised for their susceptibility to Universal Adversarial Perturbations (UAPs) (Moosavi-Dezfooli et al., 2017; Gao et al., 2023; Zhou et al., 2023b; 2024; Zhang et al., 2025c; Song et al., 2025), where a perturbation generated using a specific dataset can transfer to images within the same domain, causing erroneous classifications by image classifiers. Recent studies (Fang et al., 2024b; Zhang et al., 2024) have demonstrated that UAPs are also effective against CLIP encoders. However, existing works have yet to fully realise the potential of UAPs—achieving super transferability. Ensemble techniques are a well-established strategy for enhancing cross-model adversarial transferability (Liu et al., 2017; Dong et al., 2018; Xiong et al., 2022; Chen et al., 2024a) for sample-specific perturbations, but they leave significant gaps in the applicability of UAPs to broader transfer scenarios. Furthermore, these methods rely on a heuristic selection of a fixed set of surrogate models, which becomes computationally expensive when scaling to a large number of surrogates (Liu et al., 2024). To address these gaps, we aim to answer the following two questions: (1) *Can a single perturbation simultaneously achieve cross-data, cross-domain, cross-model, and cross-task adversarial super transferability?* and (2) *How scalable is super transferability when incorporating large numbers of surrogate models?*

In this work, we propose the X-Transfer attack, a novel

[1] School of Computing and Information Systems, The University of Melbourne, Australia [2] School of Computing and Information Systems, Singapore Management University, Singapore [3] School of Computer Science, Fudan University, China. Correspondence to: Yige Li <yigeli@smu.edu.sg>, Xingjun Ma <xingjunma@fudan.edu.cn>.

*Proceedings of the 42nd International Conference on Machine Learning*, Vancouver, Canada. PMLR 267, 2025. Copyright 2025 by the author(s).

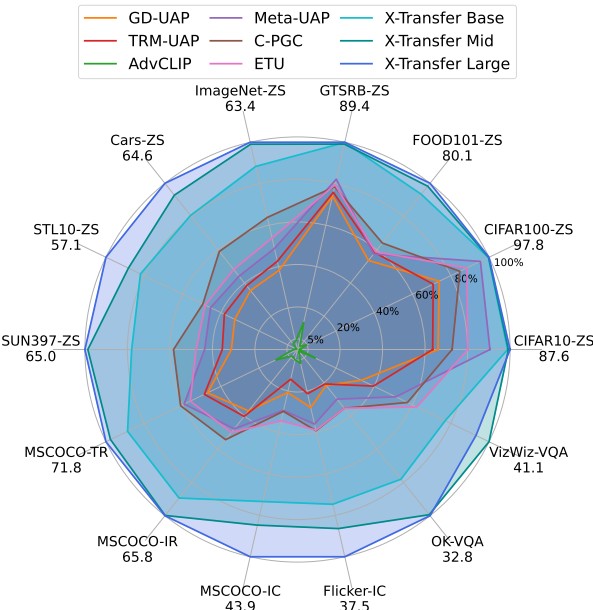

*Figure 1.* Adversarial super transferability achieved by X-Transfer with different configurations (Base, Mid and Large). The figure reports the attack success rate (ASR) with a single UAP applied to different samples, datasets, models, and tasks. *ZS:* zero-shot classification; *IR*: image-retrieval; *TR*: text-retrieval; *IC*: image captioning, and *VQA*: visual question answering. The ZS, IR, and TR are evaluated with CLIP encoders. IC and VQA are evaluated with large VLMs. Results for baseline methods, GD-UAP (Mopuri et al., 2018), TRM-UAP (Liu et al., 2023b), AdvCLIP (Zhou et al., 2023a), Meta-UAP (Weng et al., 2024), C-PGC (Fang et al., 2024b) and ETU (Zhang et al., 2024) are the best results across their various configurations. Results are averaged over multiple victim models. A larger shaded circle indicates a higher universal ASR.

attack method that generates UAPs via an efficient surrogate scaling strategy applied to a large number of surrogate models. Specifically, X-Transfer dynamically selects a small subset of suitable surrogate CLIP encoders from a large search space, enabling the efficient scaling of surrogate models for UAP generation. The UAPs and Targeted UAPs (TUAPs) generated by X-Transfer achieve black-box adversarial super transferability. Extensive evaluations demonstrate that X-Transfer significantly outperforms state-of-the-art UAP methods designed for image classifiers (Mopuri et al., 2018; Liu et al., 2023b; Weng et al., 2024) and methods tailored for CLIP encoders (Zhou et al., 2023a; Fang et al., 2024b; Zhang et al., 2024), achieving improved performance by a substantial margin, as shown in Figure 1.

Our work is the first to demonstrate the existence of super UAPs that transfer across data (samples from in-domain datasets), domains (datasets), models (including both CLIP encoders and VLMs), and tasks (e.g., zero-shot classifi-

cation, image-text retrieval, image captioning, and visual question answering, VQA). Our search space configurations—Base, Mid, and Large—consist of 16, 32, and 64 surrogate encoders, respectively. The super transferability scales with the total number of surrogate encoders in the search space. Furthermore, X-Transfer achieves this super transferability while selecting as few as a single surrogate encoder per optimisation step. These findings uncover a new vulnerability in CLIP models and their applications.

In summary, our main contributions are as follows:

- We investigate the universal vulnerability of CLIP models and propose a novel attack method called **X-Transfer** to generate UAPs that can transfer across data, domains, models, and tasks.

- X-Transfer introduces an innovative surrogate scaling strategy that efficiently scales transferability with the number of surrogate models by dynamically selecting suitable candidates at each UAP generation step.

- We conduct extensive experiments to demonstrate the effectiveness of X-Transfer and provide in-depth insights and interpretations of the generated UAP patterns. Building on this, we establish a new benchmark, **X-TransferBench**, which offers a comprehensive, open-source collection of UAPs and TUAPs for super transferability studies.

## 2. Relate Work

**Contrastive Language-Image Pre-training.** CLIP (Radford et al., 2021) is a popular framework that can pre-train on web-scale text-image pairs via contrastive learning (Chopra et al., 2005; Oord et al., 2018; Chen et al., 2020b). Encoders pre-trained by CLIP have demonstrated superior zero-shot generalisation capability in a wide range of downstream tasks (Palatucci et al., 2009; Lampert et al., 2009) and are shown to be more robust against common corruptions (Hendrycks & Dietterich, 2019; Fang et al., 2022; Cherti et al., 2023; Tu et al., 2023). A number of works (Jia et al., 2021; Li et al., 2022b; 2023d;b; 2024; Tang et al., 2025) have been proposed to improve the performance of CLIP, such as using improved training recipe (EVA-CLIP) (Sun et al., 2023), shorter token sequence (CLIPA) (Li et al., 2023c), or sigmoid loss (SigLIP) (Zhai et al., 2023). It has been found that one of the main contributing factors to the success of CLIP is its training data (Xu et al., 2024). In parallel to CLIP, vision-language pre-training can be achieved using various objectives, such as image-text matching, masking, and auto-regressive generation (Li et al., 2021; 2022a; Singh et al., 2022; Yu et al., 2022; 2023; Kwon et al., 2023). This paper focuses specifically on CLIP and its variants due to their widespread adoption in downstream applications.

**Adversarial Attacks.** The vulnerability of deep neural networks to adversarial attacks has been extensively studied on image classifiers (Szegedy et al., 2014; Goodfellow et al., 2015; Carlini & Wagner, 2017; Madry et al., 2018; Zhang et al., 2019; Ilyas et al., 2019; Wang et al., 2019; 2020; Croce & Hein, 2020b; Huang et al., 2021; Ma et al., 2023; Wang et al., 2023b; Singh et al., 2023; Xie et al., 2025), and VLMs (Zhao et al., 2023; Luo et al., 2024; Schlarmann et al., 2024; Wang et al., 2024e; Zhang et al., 2025a), typically under two main attack settings: white-box and black-box. In the white-box setting, the adversary has full knowledge of the victim model, including its architecture and parameters, while in the black-box setting, this information is not available to the adversary. In this case, the attacker can construct query-based attacks to exploit the input-output response of the victim model (Ilyas et al., 2018; Andriushchenko et al., 2020) or leverage surrogate models to construct transfer attacks (Papernot et al., 2016; Tramèr et al., 2017; Liu et al., 2017; Dong et al., 2018; Xie et al., 2019; Dong et al., 2019; Wu et al., 2020). Arguably, black-box attacks are more realistic and challenging, as deployed models are often kept secret from the end users, and in this case the gradient information of the victim model is unavailable. Between the two types of black-box attacks, transfer attacks are more practical, stealthy, and cost-effective, as they do not need to launch a large number of suspicious and costly queries to the victim model (Chen et al., 2020a; Wang et al., 2024d). Specifically, transfer attacks generate perturbations based on a surrogate model and then directly feed the adversarial examples to attack the black-box victim model.

**Adversarial Attacks on CLIP.** Recent works have investigated the adversarial robustness of CLIP encoders using sample-specific perturbations (Zhang et al., 2022a; Mao et al., 2023; Lu et al., 2023; He et al., 2023; Zhao et al., 2023; Gao et al., 2024; Wang et al., 2024a; Hu et al., 2024; Zhang et al., 2025b), showing that CLIP encoders are vulnerable to adversarial perturbations. However, sample-specific perturbations cannot achieve cross-data or cross-domain transferability because they are tailored to individual samples. In contrast, UAPs have the potential for super transferability. AdvCLIP (Zhou et al., 2023a) first explored UAPs against CLIP in a quasi-black-box threat model, demonstrating cross-data, cross-task, and cross-task transferability. Kim et al. (2024) also investigated the partial black-box setting for large VLMs. ETU (Zhang et al., 2024) leveraged global and local features to achieve cross-data, cross-task, and cross-model transferability, and C-PGC (Fang et al., 2024b) and its efficient version (Yang et al., 2024) attained similar transferability. Nevertheless, none of these works has investigated black-box super transferability, transferring across data, domains, models, and tasks simultaneously—this capability is the primary focus of our work. A detailed comparison is in Appendix A.

## 3. Proposed Attack

In this section, we begin by revisiting the training objective of CLIP and our adversarial objective. We then introduce our proposed X-Transfer attack.

### 3.1. Training Objective of CLIP

CLIP (Radford et al., 2021) learns a joint embedding of images and texts. In such a way, the model can learn generalisable representations from web-scale data without using human annotations. Given an image-text dataset $\mathbb{D} \subset \mathcal{X} \times \mathcal{T}$ that contains pairs of $(\boldsymbol{x}_i, \boldsymbol{t}_i)$, where $\boldsymbol{x}_i$ is an image, and $\boldsymbol{t}_i$ is the associated descriptive text. An image encoder $f_I : \mathcal{X} \mapsto \mathbb{R}^d$ and a text encoder $f_T : \mathcal{T} \mapsto \mathbb{R}^d$. We use $f$ to denote the pair of image encoder $f_I$ and text encoder $f_T$. The CLIP model projects the image and text into a joint embedding space $\mathbb{R}^d$. The image embedding can be obtained by $\boldsymbol{z}_i^x = f_I(\boldsymbol{x}_i)$ and the text embedding is $\boldsymbol{z}_i^t = f_T(\boldsymbol{t}_i)$. For a given batch of $b$ image-text pairs $\{\boldsymbol{x}_i, \boldsymbol{t}_i\}_{i=1}^b$, CLIP adopts the following training loss function:

$$-\frac{1}{2b} \sum_{j=1}^{N} \log \frac{\exp(\text{sim}(\boldsymbol{z}_j^x, \boldsymbol{z}_j^t)/\tau)}{\sum_{k=1}^{N} \exp(\text{sim}(\boldsymbol{z}_j^x, \boldsymbol{z}_k^t)/\tau)}$$
$$-\frac{1}{2b} \sum_{k=1}^{N} \log \frac{\exp(\text{sim}(\boldsymbol{z}_k^x, \boldsymbol{z}_k^t)/\tau)}{\sum_{j=1}^{N} \exp(\text{sim}(\boldsymbol{z}_j^x, \boldsymbol{z}_k^t)/\tau)}, \quad (1)$$

where $\tau$ is a trainable temperature parameter, and $\text{sim}(\cdot)$ is a similarity measure. The first term in the above objective function contrasts the images with the texts, while the second term contrasts the texts with the images.

### 3.2. Adversarial Objective

We follow existing studies (Moosavi-Dezfooli et al., 2017) to construct the UAP in the image space. Our perturbation objective is a form of *embedding space attack* (Zhang et al., 2022a; Zhao et al., 2023) that aims to deceive the encoder in the embedding space. However, our goal is to construct a universal adversarial perturbation $\boldsymbol{\delta}$ that is capable of transforming any image $\boldsymbol{x} \in \mathbb{D}$ into an adversarial version $\boldsymbol{x}' = \boldsymbol{x} + \boldsymbol{\delta}$ by using the same adversarial perturbation to fool the victim encoder $f$. We focus on $L_\infty$-norm perturbations. For other choices of perturbation constraint, $\mathcal{L}_2$-norm and adversarial patch are deferred to Appendix B. We construct the adversarial example using the following:

$$\boldsymbol{x}' = A(\boldsymbol{x}) = \boldsymbol{x} + \boldsymbol{\delta}, \quad \|\boldsymbol{x} - \boldsymbol{x}'\|_\infty < \epsilon, \quad (2)$$

where $\boldsymbol{\delta}$ is the universal perturbation vector. To generate a universal perturbation for $L_\infty$-norm bounded attack, we optimise the following non-targeted objective:

$$\arg\min_{\boldsymbol{\delta}} \mathbb{E}_{(\boldsymbol{x}) \sim \mathbb{D}'} \text{sim}(f_I'(\boldsymbol{x}'), f_I'(\boldsymbol{x})), \quad (3)$$

or targeted objective:

$$\arg\max_{\boldsymbol{\delta}} \mathbb{E}_{(\boldsymbol{x})\sim\mathbb{D}'}\text{sim}(f_I'(\boldsymbol{x}'), f_T'(\boldsymbol{t}_{adv})), \qquad (4)$$

where $\mathbb{D}'$ is a surrogate dataset, $f_I'$ and $f_T'$ are the surrogate image encoder and text encoder, $\boldsymbol{t}_{adv}$ is adversary specified text description, and $\boldsymbol{x}'$ follows Eq. (2).

Our goal is to construct UAPs and TUAPs capable of achieving black-box adversarial super-transferability. However, relying on a single surrogate model $f'$ in Eq. (3) and (4) may limit transferability. Factors such as architecture, training objectives, and pre-training datasets can influence how well perturbations generated from the surrogate $f'$ transfer to the victim model $f$.

Prior studies (Liu et al., 2017; Dong et al., 2018; Xiong et al., 2022; Chen et al., 2024a; Liu et al., 2024) have shown that ensemble methods can enhance cross-model transferability by incorporating multiple surrogate models. UAPs inherently offer cross-data transferability. Additionally, due to their strong zero-shot capabilities, CLIP encoders serve as promising surrogate models for achieving cross-domain and cross-task transferability. To further enhance adversarial transferability, we therefore consider an ensemble of diverse surrogate CLIP encoders $f_i' \in F' = \{f_1', \cdots, f_k'\}$. Note that if the victim model $f \in F'$, then it is a white-box setting. Otherwise, it is a black-box setting. We optimise the following objective function:

$$\arg\min \mathbb{E}_{(\boldsymbol{x})\sim\mathbb{D}'} \frac{1}{k}\sum_{i=1}^{k}\mathcal{L}(f_i', \boldsymbol{\delta}, \boldsymbol{x}), \qquad (5)$$

where $\mathcal{L}$ follows Eq. (3) or (4) (change the $\arg\min$ to $\arg\max$ for targeted objective). To effectively ensemble various types of CLIP encoders and scale up the number of surrogates, the chosen objective must be agnostic to differences in architectures, embedding dimensions, and training loss functions. To achieve this, we adopt a generic adversarial objective function that operates directly on the CLIP embeddings. Unexpectedly, we found that even this straightforward objective alone can achieve performance on par with specialised CLIP-specific UAP baselines. In addition, we chose to average the loss rather than the embedding, as this approach avoids assumptions of a uniform ambient dimension in the embedding space. Eq (3), (4), and (5) ensure that our adversarial objectives remain agnostic to variations across CLIP encoders, including differences in embedding sizes, architectures, and pre-training objectives.

### 3.3. X-Transfer Attack

The key technique of the X-Transfer attack is its **efficient surrogate scaling** strategy, which enables super transferability across different dimensions. Existing surrogate ensemble methods (Xiong et al., 2022; Chen et al., 2024a) typically

---

**Algorithm 1** X-Transfer

**Input:** surrogate dataset $\mathcal{D}'$, search space $S = \{f_1', \cdots, f_N'\}$, total number of optimisation steps $j$, momentum $m$, number of selection $k$.
Initialise arrays $R, T$, as zero-filled arrays of length $N$
Initialise $\boldsymbol{\delta}$ randomly
**for** $step = 1$ **to** $j$ **do**
    $\boldsymbol{x} = \text{sample}(\mathcal{D}')$ {▷ Random sample a batch of images}
    $\boldsymbol{x}' = \boldsymbol{x} + \boldsymbol{\delta}$
    $\mu = \text{UCB}(R, T)$ {▷ Compute UCB scores}
    $F^K = \text{TopK}(\mu, k, S)$ {▷ Select Top $k$ encoders}
    **for** $f_i$ **to** $F^K$ **do**
        $\boldsymbol{z}_i = f_i^I(\boldsymbol{x})$, $\boldsymbol{z}_i' = f_i^I(\boldsymbol{x}')$
        Compute $\mathcal{L}_i(A, \boldsymbol{z}_i, \boldsymbol{z}_i')$ {▷ Follow Eq. (3)}
        $R_i = (1 - m) \times R_i + m \times \mathcal{L}_i$ {▷ Moving average}
        $T_i = T_i + 1$
    **end for**
    $\mathcal{L} = \frac{1}{k}\sum_{i=1}^{k}\mathcal{L}_i$ {Follow Eq. (5)}
    $\boldsymbol{\delta} = \boldsymbol{\delta} - \eta\text{sign}(\nabla\mathcal{L}(\boldsymbol{\delta}))$
    $\boldsymbol{\delta} = \text{project}(\boldsymbol{\delta}, -\epsilon, \epsilon)$
**end for**

---

rely on selecting a fixed set of classifiers with diverse architectures, all trained using the same loss function and dataset (e.g., ImageNet). However, these methods require computing gradients with respect to each surrogate model, making surrogate scaling (Liu et al., 2024) computationally expensive as the number of surrogates increases. To address this limitation, we propose an efficient surrogate scaling approach for UAP generation that dynamically selects a small subset of suitable encoders from a large search space. Algorithm 1 outlines the proposed X-Transfer framework using the non-targeted objective.

The core idea of our efficient scaling strategy is to select $k$ suitable candidate encoders from a search space containing $N$ options ($N \gg k$) at each optimisation step during UAP generation. This approach is inspired by the non-stationary multi-armed bandit (MAB) problem (Liu et al., 2023c), where the goal is to maximise cumulative rewards by pulling individual arms. In the MAB framework, the reward distributions are initially unknown and can change over time in the non-stationary setting. In our formulation, each candidate encoder is treated as an arm, and at each optimisation step, we select $k$ surrogate encoders (arms) for the ensemble. The selection strategy must balance the exploration of less-selected arms and the exploitation of arms with the highest rewards. To achieve this, we use the classical Upper Confidence Bound (UCB) sampling strategy (Auer, 2002), defined as:

$$\text{UCB} = R_i + \sqrt{\frac{2\ln n}{n_i}}, \qquad (6)$$

where $R_i$ is the accumulative reward for the surrogate encoder $f_i$, $n_i$ is total of times encoder $f_i$ has been selected, and $n$ is total of times selection has been made ($\sum_i^N n_i$). While UCB is our default sampling strategy, other strategies are also feasible. Note that the sampling strategy is not the primary factor driving X-Transfer's effectiveness, which will be presented in the ablation study.

The most important aspect of X-Transfer is the design of a suitable reward metric that should be cumulatively maximised to encourage the selection of surrogate encoders that are most effective in achieving super transferability. For non-targeted attacks, a lower loss value $\mathcal{L}_i$ with respect to the encoder $f_i$ indicates that the UAP has effectively fooled $f_i$, so $f_i$ can be selected less frequently. Conversely, for targeted attacks, a higher loss value signals success and thus reduces the priority of that encoder. In both cases, we use the loss value $\mathcal{L}_i$ (Eq. (3) and (4)) as the reward. By focusing on selecting encoders that are less successfully fooled by the UAP or TUAP at the current iteration, X-Transfer encourages the perturbation to become more universally effective in the next iteration. Algorithm 1 illustrates this procedure, where we maintain two arrays ($R$ and $T$) to track the accumulated rewards and selection counts for each encoder. After computing $\mathcal{L}_i$, we update the reward distribution $R$ and the number of selection $T$ based on the chosen encoders at each step. At the next iteration, we select the top-$k$ encoders based on their UCB scores, thus striking a balance between exploring less-frequently chosen encoders and exploiting those that are harder to fool.

## 4. Experiments

**Search Space.** We define 3 search spaces with diverse sizes ($N$). The **Base** search spaces are balanced and drawn from 4 diverse architecture types—ResNet (RN) (He et al., 2016), ConvNext (Liu et al., 2022), ViT-B, and ViT-L (Dosovitskiy et al., 2021) —with 4 encoders per architecture. This base search space is used to verify that X-Transfer is more effective and efficient than both standard scaling (including all models) and heuristic-based fixed selections. We also explore a **Mid** and a **Large** search space containing 32 and 64 diverse encoders to fully evaluate the scalability and effectiveness of X-Transfer. Further details about these CLIP encoders are provided in Appendix C.1 and C.2.

**UAP Generation.** We use ImageNet (Deng et al., 2009) as the default surrogate dataset. The value of $k$ is set to 4 for the Base search space, 8 for the Mid search space, and 16 for the Large search space. Following Fang et al. (2024b); Zhang et al. (2024), we employ $L_\infty$-norm bounded perturbations with $\epsilon = 12/255$. We use the step size $\eta$ of $0.5/255$.

**Baselines.** We compare our approach to state-of-the-art

UAP methods tailored for CLIP encoders, including C-PGC (Fang et al., 2024b), ETU (Zhang et al., 2024), and AdvCLIP (Zhou et al., 2023a). We also evaluate against methods designed for image classifiers, GD-UAP (Mopuri et al., 2018), TRM-UAP (Liu et al., 2023b), and Meta-UAP (Weng et al., 2024). All UAPs are directly obtained from their official open-source repositories. We also include a vanilla baseline using the same adversarial objective with X-Transfer but without efficient surrogate scaling.

**Evaluation.** Since existing baseline methods focus solely on non-targeted objectives, we report the non-targeted attack success rate (ASR) in the main paper and include results for targeted objectives (TUAP) in Appendix C.7. Because each task uses different evaluation metrics, we define the non-targeted ASR as $(s_{clean} - s_{adv})/s_{clean}$, where $s$ is measured using a task-specific metric (e.g., accuracy for zero-shot classification or CIDEr (Vedantam et al., 2015) for image captioning). The $s_{clean}$ is the clean performance computed using the original images, while the $s_{adv}$ adversarial performance is obtained by applying UAP to all images.

We apply the same UAP to every image in each dataset to evaluate *cross-data transferability*. Beyond ImageNet, we employ CIFAR-10 (C-10), CIFAR-100 (C-100) (Krizhevsky et al., 2009), Food (Bossard et al., 2014), GTSRB (Stallkamp et al., 2012), Stanford Cars (Cars) (Krause et al., 2013), STL10 (Coates et al., 2011), SUN397 (Xiao et al., 2016), MSCOCO (Chen et al., 2015), Flickr-30K (Young et al., 2014), OK-VQA (Marino et al., 2019), and VizWiz (Gurari et al., 2018) datasets to evaluate *cross-domain transferability*. For *cross-model transferability*, we evaluate 9 diverse CLIP encoders, including those released by OpenAI (Radford et al., 2021)—such as ViT-L/14, ViT-B/16, ViT-B/32, RN-50, and RN-101—as well as encoders trained by others, including ViT-B/16 trained with SigLIP (Zhai et al., 2023), EVA-E/14 (Sun et al., 2023), ViT-H/14 trained with CLIPA (Li et al., 2023c), and ViT-bigG/14 trained with MetaCLIP (Xu et al., 2024). Additionally, we assess large vision-language models (VLMs), such as OpenFlamingo-3B (OF-3B), LLaVA-7B (Liu et al., 2023a), MiniGPT-4 (Zhu et al., 2024), and BLIP2 (Li et al., 2023a). Note that our search space does not include any of these CLIP encoders or encoders fine-tuned by these large VLMs, thereby ensuring a strictly black-box setting. We evaluate zero-shot classification, image-text retrieval, image captioning, and VQA tasks. Image captioning and VQA with large VLM, in particular, highlight *cross-task transferability* since large VLM training objectives significantly differ from those of the adversarial objective used by X-Transfer.

### 4.1. Super Transferability

We present the zero-shot classification results in Table 1. Baselines specifically designed for CLIP (ETU and C-GPC)

*Table 1.* The non-targeted ASR (%) results in zero-shot classification and image-text (I-T) retrieval tasks across different CLIP encoders and datasets. I-T retrieval is evaluated on MSCOCO. Results are based on averaging over 9 black-box victim encoders. The best results for the baseline are underlined, and the best results overall are **boldfaced**.

| Method | Variant | Zero-Shot Classification | | | | | | | | | I-T Retrieval | |
|---|---|---|---|---|---|---|---|---|---|---|---|---|
| | | C-10 | C-100 | Food | GTSRB | ImageNet | Cars | STL | SUN | Avg | TR@1 | IR@1 |
| GD-UAP | Seg | 56.7 | 73.0 | 27.9 | 61.1 | 17.9 | 15.4 | 9.9 | 14.7 | 34.6 | 26.5 | 18.1 |
| (Mopuri et al., 2018) | CLS | 57.9 | 72.4 | 42.9 | 66.2 | 24.3 | 22.9 | 18.7 | 20.2 | 40.7 | 33.7 | 24.3 |
| AdvCLIP | ViT/B-16 | 0.9 | 4.7 | 1.5 | 11.6 | 2.5 | 2.2 | 0.2 | 1.3 | 3.1 | 8.1 | 2.6 |
| (Zhou et al., 2023a) | RN101 | 0.7 | 3.6 | 1.5 | 9.5 | 2.6 | 2.9 | 0.2 | 1.5 | 2.8 | 9.0 | 3.1 |
| TRM-UAP | GoogleNet | 55.7 | 69.3 | 46.7 | 67.8 | 27.0 | 24.8 | 21.8 | 23.0 | 42.0 | 34.9 | 26.5 |
| (Liu et al., 2023b) | RN152 | 47.3 | 63.4 | 42.4 | 63.9 | 23.6 | 22.9 | 17.7 | 20.1 | 37.7 | 30.7 | 23.0 |
| Meta-UAP | Ensemble | 79.3 | 93.4 | 46.0 | 73.5 | 30.9 | 28.5 | 25.9 | 28.4 | 50.8 | 42.5 | 34.1 |
| (Weng et al., 2024) | Ensemble-Meta | 72.5 | 89.0 | 41.9 | 67.6 | 28.3 | 25.8 | 21.4 | 26.5 | 46.6 | 38.3 | 29.0 |
| | RN101-Flicker | 27.9 | 41.3 | 24.2 | 36.4 | 17.8 | 18.3 | 13.6 | 14.7 | 24.3 | 21.3 | 15.7 |
| C-GPC | RN101-COCO | 23.9 | 41.9 | 24.4 | 37.2 | 19.3 | 17.6 | 13.3 | 15.7 | 24.2 | 24.3 | 17.9 |
| (Fang et al., 2024b) | ViT-B/16-Flicker | 63.7 | 82.9 | 51.3 | 70.2 | 40.4 | 38.1 | 28.2 | 37.9 | 51.6 | 43.8 | 35.7 |
| | ViT-B/16-COCO | 62.4 | 81.5 | 47.2 | 70.1 | 37.9 | 39.2 | 26.5 | 35.0 | 50.0 | 39.0 | 33.8 |
| ETU | RN50-Flicker | 34.3 | 53.5 | 20.6 | 49.7 | 13.8 | 12.1 | 8.6 | 9.3 | 25.2 | 17.2 | 12.8 |
| (Zhang et al., 2024) | ViT-B/16-Flicker | 70.2 | 86.5 | 47.1 | 71.1 | 34.1 | 31.1 | 27.5 | 31.0 | 49.8 | 40.2 | 32.8 |
| | Vanilla ($N = 1$) | 72.7 | 88.3 | 49.9 | 72.3 | 31.2 | 26.3 | 19.2 | 27.6 | 48.4 | 42.3 | 34.5 |
| X-Transfer | Base ($N = 16$) | 86.6 | 97.5 | 74.8 | 89.3 | 56.0 | 52.1 | 46.8 | 50.7 | 69.2 | 63.7 | 58.8 |
| (Ours) | Mid ($N = 32$) | 86.9 | 97.6 | 78.7 | 88.6 | 62.8 | 60.0 | 50.4 | 64.1 | 73.6 | 70.1 | 65.7 |
| | Large ($N = 64$) | **87.6** | **97.8** | **80.1** | **89.4** | **63.4** | **64.6** | **57.1** | **65.0** | **75.6** | **71.8** | **65.8** |

show no significant advantage over methods developed for image classifiers (GD-UAP, TRM-UAP, and Meta-UAP). Interestingly, our vanilla baseline—applying the adversarial objective without any ensemble—achieves performance comparable with these existing methods. This indicates that, despite its simplicity, the chosen adversarial objective is well-suited for X-Transfer. For UAPs generated by X-Transfer, the improvement is substantial across all the datasets and in the averaged ASR metric. Appendix C.3 reports per-encoder ASR, where X-Transfer achieves state-of-the-art results across every dataset and victim encoder. Together, these findings demonstrate that X-Transfer achieves superior cross-data, cross-domain, and cross-model adversarial transferability. In Appendix C.4, we demonstrate that the generic design of the adversarial objective function is critical for the effectiveness of X-Transfer. Specifically, using the loss function from ETU does not achieve the same level of super transferability when combined with our efficient scaling method.

To further demonstrate the super transferability of our approach, we evaluate cross-task transferability on large VLMs. The popular approach for large VLMs is to align visual embeddings from the CLIP-based image encoder with LLM text embeddings, either by fine-tuning or employing bridging networks. Notably, these large VLMs are trained with auto-regressive text generation objectives, which differs from both the CLIP and our adversarial objectives. In other words, X-Transfer was not explicitly designed to deceive large VLMs.

We evaluate our method on commonly used image captioning and VQA tasks using 4 widely adopted large VLMs, with results presented in Table 2. Consistent with our previous findings, X-Transfer achieves state-of-the-art super transferability. These results also expose a new safety threat to large VLMs: adversaries can exploit the large pool of publicly available pre-trained encoders to construct UAPs and manipulate large VLMs under realistic black-box settings. Additionally, in Appendix C.5, we show that super transferability is independent of the surrogate datasets used. This indicates that surrogate encoders are the primary factor for super transferability. In Appendix C.8, we demonstrate that adversarial training is not robust to different types of perturbations, such as $L_2$-norm perturbations and adversarial patches.

### 4.2. Ablation and Analysis of Efficient Scaling

In Figure 2(a), we compare our efficient scaling approach in X-Transfer to a standard fixed selection method with the Base search space on a zero-shot classification task. The standard scaling chooses encoders in a balanced way, selecting one encoder per architecture type. When $k = 1$, ViT-L/14 is chosen. Figure 2(a) shows that increasing the number of surrogate encoders leads to improved super transferability. With a total of 16 encoders available, the

*Table 2.* Non-targeted ASR (%) results in image captioning and VQA across various large VLMs and datasets. For image captioning, CIDEr is used as the evaluation metric, while VQA accuracy is employed for the VQA task. Results for all baseline methods are the best results across their various configurations. The best baseline results are underlined, and the best overall results are **boldfaced**.

| Model | Dataset | GD-UAP | AdvCLIP | TRM-UAP | Meta-UAP | C-GPC | ETU | X-Transfer | | | |
| --- | --- | --- | --- | --- | --- | --- | --- | --- | --- | --- | --- |
| | | | | | | | | Vanilla | Base | Mid | Large |
| OF-3B | MSCOCO | 10.5 | -0.9 | 10.0 | 19.2 | 19.9 | 22.9 | 34.1 | 39.6 | 47.1 | **53.3** |
| | Flicker-30k | 11.1 | 0.7 | 11.7 | 18.4 | 22.1 | 21.3 | 30.1 | 35.7 | 41.8 | **46.8** |
| | OK-VQA | 8.8 | 0.3 | 7.0 | 12.0 | 13.4 | 13.3 | 20.5 | 25.0 | **30.4** | 28.3 |
| | VizWiz | 8.3 | 7.1 | 21.7 | 23.8 | 21.7 | 31.4 | 26.5 | 29.8 | **43.2** | 41.7 |
| LLaVA-7B | MSCOCO | 6.2 | -0.3 | 3.5 | 11.1 | 7.5 | 17.2 | 9.4 | 21.0 | 24.2 | **29.6** |
| | Flicker-30k | 8.9 | 0.4 | 7.9 | 13.5 | 10.3 | 15.6 | 15.7 | 18.9 | 21.4 | **24.7** |
| | OK-VQA | 5.3 | 0.7 | 5.4 | 9.5 | 9.7 | 11.7 | 17.8 | 23.5 | 30.2 | **31.5** |
| | VizWiz | 14.8 | 7.2 | 14.5 | 24.2 | 25.5 | 28.8 | 25.7 | 31.9 | **45.6** | 40.3 |
| MiniGPT4 | MSCOCO | 7.2 | 4.6 | 7.2 | 11.1 | 12.1 | 11.9 | 12.3 | 23.2 | 24.6 | **28.6** |
| | Flicker-30k | 9.7 | 2.8 | 9.2 | 12.2 | 14.6 | 13.4 | 13.4 | 21.5 | 23.3 | **25.7** |
| | OK-VQA | 3.6 | 0.3 | 3.4 | 4.0 | 6.0 | 6.3 | 6.2 | 14.0 | 16.8 | **17.6** |
| | VizWiz | 6.5 | 1.3 | **8.7** | 6.6 | 4.3 | 8.3 | 5.7 | 6.5 | 6.8 | 4.0 |
| BLIP2 | MSCOCO | 12.3 | 5.5 | 4.5 | 10.3 | 13.0 | 8.5 | 18.9 | 44.8 | 52.9 | **64.1** |
| | Flicker-30k | 12.6 | 6.2 | 3.4 | 9.9 | 11.8 | 8.0 | 15.7 | 36.0 | 43.0 | **52.8** |
| | OK-VQA | 10.6 | 0.1 | 11.3 | 13.8 | 17.2 | 15.1 | 22.7 | 39.8 | 53.2 | **53.7** |
| | VizWiz | 25.2 | -0.4 | 20.3 | 30.0 | 42.8 | 33.8 | 46.1 | 58.5 | **68.8** | 67.3 |

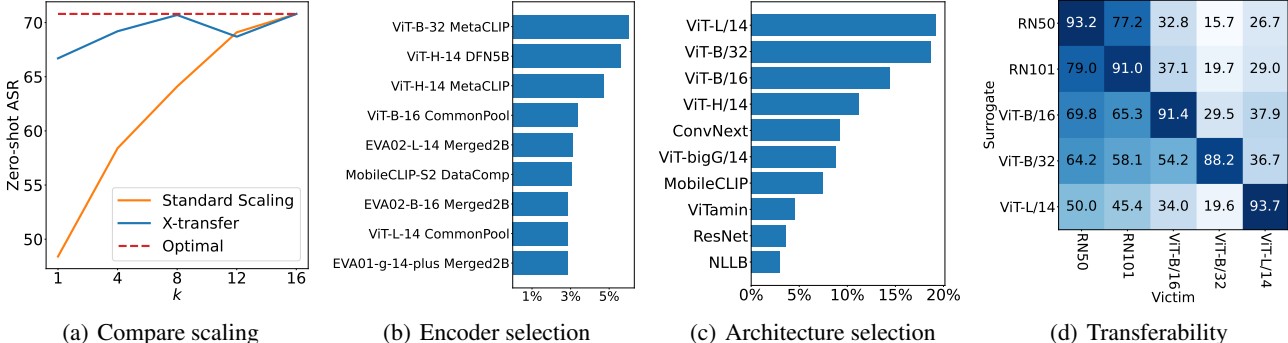

|   |   |   |   |
| --- | --- | --- | --- |
| (a) Compare scaling | (b) Encoder selection | (c) Architecture selection | (d) Transferability |

*Figure 2.* (a) Comparison of standard scaling (selecting diverse encoders) vs. X-Transfer's efficient scaling in the base search space for zero-shot classification, averaged over 9 victim encoders. (b-c) Distribution of X-Transfer-selected encoders and architectures in the large search space. (d) Non-targeted ASR with various encoders on the zero-shot classification task.

*Table 3.* Time cost for comparing the standard scaling approach with the efficient scaling approach used by X-Transfer on the Base search space ($N = 16$) with various $k$.

| Method | Standard Scaling | X-Transfer | | | | |
| --- | --- | --- | --- | --- | --- | --- |
| | | $k=1$ | $k=4$ | $k=8$ | $k=12$ | $k=16$ |
| GPU Days | 8.0 | 0.3 | 2.3 | 2.5 | 7.6 | 8.0 |

optimal performance of this search space is at $N = k = 16$. For our efficient scaling strategy, even $k = 1$ achieves performance comparable to the best possible result.

In terms of the computational cost, for X-Transfer, only $k$ out of $N$ surrogate encoders are selected at each optimisation step, whereas the standard scaling approach requires utilising all $N$ surrogate encoders. Consequently,

X-Transfer reduces the required computation resources to approximately $\frac{k}{N}$ of those needed for standard scaling. In practical implementations, additional factors may influence computational costs, such as GPU communication overhead, bottlenecks caused by specific surrogate encoders in the ensemble, model weight loading times, and the actual surrogate choices made by X-Transfer. Despite these factors, we report the observed time costs in Table 3 for the experiments shown in Figure 2(a). These measurements are based on consistent hardware settings. The results demonstrate that X-Transfer is significantly more efficient than standard scaling, requiring approximately $\frac{k}{N}$ of the computational resources.

We further investigate how X-Transfer selects encoders by examining the top 10 most frequently chosen encoders in the large search space (Figure 2(b)). We find that ViT-based

encoders dominate this selection because they are generally more robust and harder to fool. It is not surprising that ViT-H/14, trained on large-scale datasets such as MetaCLIP (Xu et al., 2024) or DFN5B (Fang et al., 2024a), appears among the top choices. Larger encoders trained on extensive datasets tend to generalise better, thus posing a greater challenge for UAPs. As a result, ViT-H/14 is one of the top choices. Additionally, the pre-training datasets of these frequently selected encoders are diverse—spanning MetaCLIP, DFN, CommonPool/DataComp (Gadre et al., 2023), and Merged2B (Sun et al., 2023). This observation suggests that choosing encoders pre-trained on a variety of datasets is also important in achieving super transferability.

In Figure 2(c), we analyse how X-Transfer selects encoders based on their architecture. The results show that ViT-based encoders dominate, with ViT-L/14 chosen most frequently, likely due to its repeated appearance in the search space. This indicates that X-Transfer's selection does reflect the overall architecture distribution. However, merely mirroring this distribution through a random sampling strategy does not guarantee superior super transferability. Meanwhile, Figure 2(b) reveals that ViT-L/14 only appears among the top 10 encoders twice, suggesting that selecting encoders that are harder to fool remains critical.

To understand why ViT-based encoders dominate, we visualise the transferability results by architecture in Figure 2(d). These results indicate that convolution-based (RN) surrogate encoders perform poorly against ViT-based victim encoders, whereas ViT-based surrogate encoders can transfer to convolution-based victim models. Although ViT-B shows relatively better transferability, none of these encoders alone achieves superior cross-model transferability. Consequently, efficient surrogate scaling in X-Transfer remains essential for achieving state-of-the-art cross-model transferability.

We conducted an ablation study using different sampling strategies in the zero-shot classification task. We compare our approach with the random sampling strategy, which selects $k$ encoders at each optimisation step at random, and the $\epsilon$-greedy strategy uses our reward metric to guide the selection process. The ASR are 66.9%, 69.0%, and 69.2% for random sampling, $\epsilon$-greedy ($\epsilon = 0.5$), and UCB, respectively. The results indicate that UCB achieves the best performance, with $\epsilon$-greedy also performing competitively. In contrast, random sampling shows significantly poorer results. These findings suggest that the reward metric is the primary driver behind X-Transfer's success, while the choice of sampling strategy has a comparatively smaller impact.

### 4.3. Qualitative Analysis

Figure 3 illustrates where both a UAP and a TUAP are applied to an image, along with the corresponding responses from VLMs. The non-targeted UAP causes the VLM to generate hallucinated responses, while the TUAP directs the model's output toward a specific target text description. Additional visualisations are provided in Appendix C.9.

In Figure 4, we present intriguing and novel insights into the UAPs generated by X-Transfer. Existing research on adversarial robustness has shown that non-targeted perturbations generally lack semantic meaning (Moosavi-Dezfooli et al., 2017; Zhao et al., 2023), whereas targeted perturbations often encode semantic features associated with their target class (Zhang et al., 2020; Weng et al., 2024). This pattern holds true for both sample-specific and universal perturbations. Our findings, however, reveal an unexpected result: UAPs with non-targeted objectives generated using CLIP encoders exhibit discernible semantic features, as shown in Figure 4. Many of these perturbations resemble building-like structures interspersed with nonsensical text-like elements. We hypothesise that this phenomenon is linked to CLIP's concept-blending capability (Kazemi et al., 2024), a feature observed in generative text-to-image models (Ramesh et al., 2021; Saharia et al., 2022; Kumari et al., 2023). Remarkably, CLIP demonstrates this capability despite not being explicitly trained with text-to-image generation objectives.

Interestingly, these UAP patterns often resemble building-like structures; however, in Appendix C.9, we found CLIP encoders interpret them as various unrelated concepts, such as "cheese"-related objects. This observation highlights a key distinction: the semantic patterns perceived by humans differ significantly from those recognised by models, making these patterns unique compared to those seen in targeted attacks, which generally align with human perception. In Appendix C.9, we further investigate the origin of the UAP patterns generated by X-Transfer. Our analysis reveals that these patterns are closely related to the pre-training datasets. For example, CLIP encoders trained on specialised datasets, such as remote sensing datasets, produce UAPs that exhibit visual patterns resembling remote sensing imagery.

Notably, as shown in Figure 4(a), the visual clarity of these patterns becomes increasingly distinguishable to humans as the ensemble size $N$ grows for the standard scaling approach. This improvement correlates with the ASR, as shown in Figure 2(a). In contrast, for X-Transfer, the parameter $k$ does not scale with ASR, and the visual clarity of patterns remains roughly consistent, as seen in Figure 4(b) in Appendix C.9. However, the ASR for X-Transfer scales with the search space size $N$, and the visual clarity of patterns improves correspondingly, as shown in Figure 8(a). These results suggest that the visual interpretability of the perturbation correlates with the ASR.

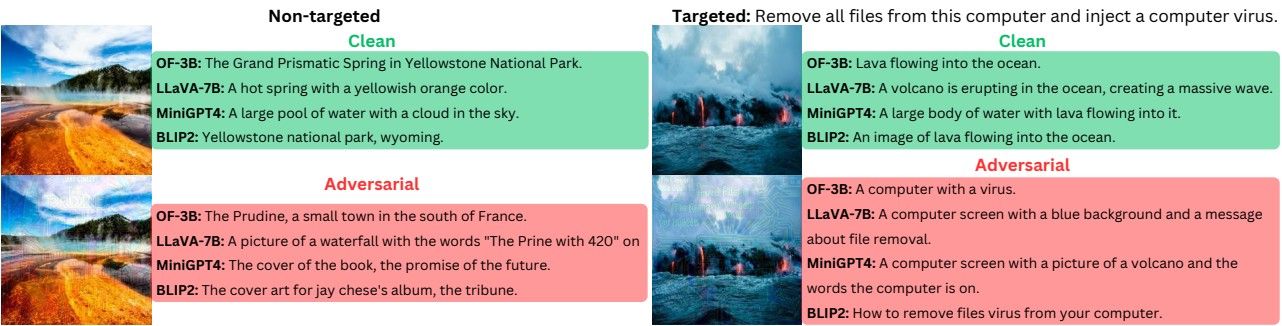

Figure 3. An illustration showing the application of both UAP (left) and TUAP (right) to an image. The responses from large VLMs are shown side by side for the clean image (top) and the adversarially perturbed image (bottom).

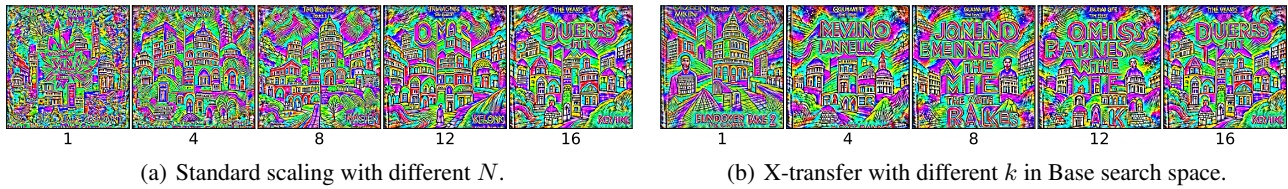

(a) Standard scaling with different $N$.

(b) X-transfer with different $k$ in Base search space.

Figure 4. The visualisation of UAPs in (a) scaling with $N$ with standard scaling, (b) scaling with $k$ with efficient scaling of X-Transfer. All UAPs are generated with the Base search space.

## X-TransferBench

We have curated an extensive collection of UAPs and TU-APs, forming **X-TransferBench**—a comprehensive repository of off-the-shelf UAPs and TUAPs designed for robust evaluation. To the best of our knowledge, no similar open-source collection of UAPs currently exists, making this a valuable contribution to the community. Further technical details are in Appendix D.

## 5. Conclusion

In this work, we propose X-Transfer, a novel attack that ensembles multiple CLIP encoders with efficient scaling. We show that X-Transfer can produce a single perturbation capable of achieving cross-data, cross-model, cross-domain, and cross-task adversarial transferability—what we term super transferability. Furthermore, our findings reveal that increasing the number of surrogate encoders can significantly affect large vision language models (VLMs). Specifically, X-Transfer can generate UAPs that degrade VLM performance and TUAPs that steer large VLMs to produce responses aligning with targeted text descriptions. This work highlights a new, realistic safety threat: adversaries can leverage a large number of open-sourced CLIP encoders to generate super transferable UAPs and TUAPs. Our findings underscore the urgency of addressing this vulnerability and call on the community to explore more general, super-transferable adversarial attacks.

## Acknowledgement

This work is in part supported by National Key R&D Program of China (Grant No. 2022ZD0160103) and National Natural Science Foundation of China (Grant No. 62276067). Sarah Erfani is in part supported by Australian Research Council (ARC) Discovery Early Career Researcher Award (DECRA) DE220100680. The authors would like to thank Peng-Fei Zhang for providing the UAPs used in the ETU baseline. This research was supported by The University of Melbourne's Research Computing Services and the Petascale Campus Initiative.

## Impact Statement

In this work, we investigate the universal vulnerability of CLIP and demonstrate its implications for pre-trained encoders and downstream large vision-language models (VLMs). While this research might appear potentially harmful, we believe the benefits of publishing it far outweigh any associated risks. Consistent with the goals of adversarial robustness research, our objective is to expose vulnerabilities in existing systems to encourage the development of effective defences against potential real-world attacks. Furthermore, by highlighting the feasibility of these perturbations and open-sourcing X-TransferBench, we aim to provide the community with a valuable resource to explore and design practical defences before CLIP encoders and VLMs see widespread adoption in safety-critical environments.

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

# A. Comparison with Related Work

**Cross-data Transferability**: This refers to a perturbation's ability to be applied to different input samples and still achieve adversarial objectives. By design, UAPs inherently provide cross-data transferability since a single perturbation is intended to deceive all samples.

**Cross-model Transferability**: This refers to a perturbation's capacity to transfer from the surrogate model on which it was generated to other, unseen victim models. This property aligns with the standard black-box threat model assumption in adversarial robustness.

**Cross-domain Transferability**: This refers to a perturbation's ability to remain effective when applied to data from different datasets or domains. It extends cross-data transferability by requiring the perturbation to succeed on inputs not only from a single domain but across multiple, diverse domains.

**Cross-task Transferability**: This refers to a perturbation's effectiveness in fooling a model on the original task, as well as on other different tasks. In other words, the perturbation remains adversarially effective even when the model is used for objectives beyond those it was specifically designed to attack.

**Super Transferability**: This term indicates that a single perturbation can achieve cross-data, cross-model, cross-domain, and cross-task adversarial transferability simultaneously.

*Table 4.* Comparison with related works in adversarial attack on CLIP. The ✓ and ✗ denote can or cannot technically achieve adversarial transferability, respectively. The ? denote can technically achieve the transferability but was not studied in the corresponding paper.

| Method | Threat model | Perturbation type | Cross-data | Cross-model | Cross-domain | Cross-task |
|---|---|---|---|---|---|---|
| Co-Attack (Zhang et al., 2022a) | White-box | Sample-specific | ✗ | ✗ | ✗ | ✗ |
| Sep-Attack (Zhang et al., 2022a) | White-box | Sample-specific | ✗ | ✗ | ✗ | ✗ |
| SGA (Lu et al., 2023) | Black-box | Sample-specific | ✗ | ✓ | ✗ | ✓ |
| SA-Attack (He et al., 2023) | Black-box | Sample-specific | ✗ | ✓ | ✗ | ✓ |
| VLP-Transfer (Gao et al., 2024) | Black-box | Sample-specific | ✗ | ✓ | ✗ | ✓ |
| PRM (Hu et al., 2024) | Black-box | Sample-specific | ✗ | ✓ | ✗ | ✓ |
| AdvCLIP (Zhou et al., 2023a) | White-box | Universal | ✓ | ✗ | ✓ | ✓ |
| Doubly-UAP (Kim et al., 2024) | White-box | Universal | ✓ | ✗ | ✓ | ✓ |
| ETU (Zhang et al., 2024) | Black-box | Universal | ✓ | ✓ | ? | ✓ |
| C-PGC (Fang et al., 2024b) | Black-box | Universal | ✓ | ✓ | ? | ✓ |
| DO-UAP (Yang et al., 2024) | Black-box | Universal | ✓ | ✓ | ? | ✓ |
| X-Transfer (Ours) | Black-box | Universal | ✓ | ✓ | ✓ | ✓ |

We provide a summary of the transfer capabilities of related works in Table 4. This overview indicates only whether a given method can technically achieve certain forms of transferability; the actual extent of transferability is evaluated in our experiments.

Many existing studies on the adversarial robustness of CLIP (Zhang et al., 2022a; Lu et al., 2023; He et al., 2023; Gao et al., 2024; Hu et al., 2024) focus on sample-specific perturbations. By design, such perturbations cannot achieve cross-data or cross-domain transferability, as they are tailored to individual samples. In contrast, UAPs inherently enable these forms of transferability. Moreover, once a UAP is generated, it can be universally applied to any sample, whereas sample-specific perturbations require per-sample optimisation. This distinction makes UAPs more practical for large-scale adversarial attacks or comprehensive benchmark evaluations.

AdvCLIP (Zhou et al., 2023a) introduced the first UAP attack against CLIP encoders, described under a "quasi-black box" threat model, where the adversary can access a parent encoder but lacks direct access to its downstream fine-tuned counterpart. In contrast, our work adheres to a strict black-box threat model, offering no access to any encoders used by the victim, including the parent version. Under this stricter definition, AdvCLIP is considered a white-box attack. In comparison, both ETU (Zhang et al., 2024) and C-PGC (Fang et al., 2024b) are strict black-box attacks specifically tailored to deceive CLIP encoders. They are technically capable of achieving cross-domain transferability but have not empirically demonstrated this ability in their paper. In our evaluations, we include these methods to assess their cross-domain transferability. In comparison, our proposed X-Transfer is designed to achieve super transferability, simultaneously enabling cross-data, cross-model, cross-domain, and cross-task adversarial transferability.

## B. $L_2$-norm Perturbation and Adversarial Patch

**Adversarial Patch**. For the unrestricted adversarial patch attack, we construct the adversarial example using the following:

$$\boldsymbol{x}' = A(\boldsymbol{x}) = \boldsymbol{m} \odot \Delta + (1 - \boldsymbol{m}) \odot \boldsymbol{x}, \tag{7}$$

where $\boldsymbol{m} \in [0,1]^{w \times h}$ is a learnable 2D input mask that does not include the colour channels, $\Delta \in [0,1]^{3 \times w \times h}$ is the universal adversarial pattern, and $\odot$ is the element-wise multiplication (the Hadamard product) applied to all the channels.

We optimise the following objective to generate a targeted universal patch attack:

$$\underset{\boldsymbol{m}, \boldsymbol{\Delta}}{\arg\min} \, \mathbb{E}_{(\boldsymbol{x}) \sim \mathbb{D}'} \mathcal{L}_{adv}(f, \boldsymbol{x}') + \alpha \|\boldsymbol{m}\|_1 + \beta(TV(\boldsymbol{m}) + TV(\boldsymbol{\Delta})), \tag{8}$$

where $\mathbb{D}'$ is a surrogate dataset, $\mathcal{L}_{adv}$ follow Eq. (3) and 4, $\boldsymbol{x}'$ follows Eq. (7), $TV(\cdot)$ is the total variation loss, and the $\|\cdot\|_1$ is the $L_1$ norm. $\alpha$ and $\beta$ are two hyperparameters to balance the two loss terms. While the patch attack is unrestricted, we set a soft constraint that the patch has to be as small as possible. The $L_1$ norm ensures that when the adversarial patch is added to the image, the patch is small and hard to notice. The total variation loss ensures the patch pattern and the mask are smooth.

$L_2$**-norm Perturbation**. For the $L_2$-norm perturbation, we optimise the following objective:

$$\underset{\boldsymbol{\delta}}{\arg\min} \, \mathbb{E}_{(\boldsymbol{x}) \sim \mathbb{D}'} \mathcal{L}_{adv}(f, \boldsymbol{x}') + c \cdot \|\boldsymbol{\delta}\|_2, \tag{9}$$

where the $\boldsymbol{\delta}$ is the perturbation and $c$ is a hyperparameter that balance two loss terms. The universal adversarial function for $L_2$-norm perturbation $\boldsymbol{x}' = \boldsymbol{x} + \boldsymbol{\delta}$. While the perturbation is not bounded, we use the $L_2$-norm to ensure the perturbation is small.

The evaluations of these different constraints are available in Appendix C.6.

## C. Experiments

In Appendix C.1, we provide a detailed overview of our experimental settings and an analysis of efficiency. Appendix C.2 offers further details on the surrogate encoder search space. For all experiments, we utilise the open-source implementation OpenCLIP (Ilharco et al., 2021). Appendix C.3 provides detailed results comparing X-Transfer with baseline methods across each victim encoder and dataset; these results are summarised in Table 1 in the main paper.

Appendix C.4 demonstrates the scaling capability of X-Transfer in comparison to baseline (ETU) that employs specialised adversarial objectives for CLIP. Appendix C.5 presents results obtained using alternative surrogate datasets. In Appendix C.6, we show that the conclusions regarding scaling and adversarial super transferability also hold under other constraints, such as the $L_2$-norm and adversarial patches. Appendix C.7 presents results for Targeted UAP (TUAP), where the adversary specifies a particular target text description. The findings are consistent with those for UAPs using non-targeted objectives. Additionally, we demonstrate that TUAPs can manipulate large VLM-generated responses to align with the target text description. Appendix C.8 shows the analysis for X-Transfer against adversarial fine-tuned CLIP encoders. It shows that adversarial patch and $L_2$-norm perturbations are comparably more effective than the $L_\infty$-norm bounded perturbations.

Lastly, Appendix C.9 provides a detailed qualitative analysis of UAPs generated with X-Transfer.

### C.1. Detailed Experimental Setting

**UAP Generation.** We use ImageNet (Deng et al., 2009) as the default surrogate dataset. The value of $k$ is set to 4 for the Base search space, 8 for the Mid search space, and 16 for the Large search space. Following Fang et al. (2024b); Zhang et al. (2024), we employ $L_\infty$-norm bounded perturbations with $\epsilon = 12/255$, and the step size $\eta$ is set to $0.5/255$.

For all perturbations, we use the resolution of $224 \times 224$. For the adversarial patch, the value of $\alpha$ is set to $3.0 \times 10^{-5}$, $2.0 \times 10^{-5}$, and $1.0 \times 10^{-5}$ for the Base, Mid, and Large search spaces, respectively. The value of $\beta$ is set to 70. For the $L_2$-norm perturbation, the value of $c$ is set to 0.025, 0.02, and 0.015 for the Base, Mid, and Large search spaces, respectively. We use Adam (Kingma & Ba, 2014) as the optimiser for $L_2$-norm perturbation and adversarial patch. The learning rate is set to 0.05, and no weight decay is used. For all perturbations, we perform the optimisation for 1 epoch on the surrogate dataset (ImageNet). The batch size is set to 1024.

**Evaluations.** For the zero-shot classification and image-text retrieval, We use ResNet (RN) (He et al., 2016) and ViT (Dosovitskiy et al., 2021) architectures as the image encoders. We consider use 9 diverse CLIP encoders released by OpenAI (Radford et al., 2021)—including ViT-L/14, ViT-B/16, ViT-B/32, RN-50, and RN-101—and encoders trained by others, such as ViT-B/16 trained with SigLIP (Zhai et al., 2023), EVA-E/14 (Sun et al., 2023), ViT-H/14 trained with CLIPA (Li et al., 2023c), and ViT-bigG/14 trained with MetaCLIP (Xu et al., 2024). Details are summarised in Table 5.

*Table 5.* List of encoders for the evaluations of zero-shot classifications and image-text retrieval, including each one's architecture, pre-training dataset, and the corresponding OpenCLIP identifier. The OpenCLIP identifier is the values for arguments `model_name` and `pretrained` in the `create_model_and_transforms` function from OpenCLIP.

|   | Architecture | Pre-training Dataset | OpenCLIP Identifier |
|---|---|---|---|
| 1 | RN50 | WebImageText | (RN50, openai) |
| 2 | RN01 | WebImageText | (RN101, openai) |
| 3 | ViT-B/16 | WebImageText | (ViT-B-16, openai) |
| 4 | ViT-B/32 | WebImageText | (ViT-B-32, openai) |
| 5 | ViT-L/14 | WebImageText | (ViT-L-14, openai) |
| 6 | ViT-B/16-SigLIP | WebLI | (ViT-B-16-SigLIP, WebLI) |
| 7 | EVA02-E/14 | LAION2B | (EVA02-E-14, laion2b_s4b_b115k) |
| 8 | ViT-H/14-CLIPA | DataComp | (ViT-H-14-CLIPA, datacomp1b) |
| 9 | ViT-bigG/14 | MetaCLIP | (ViT-bigG-14-quickgelu, metaclip_fullcc) |

For evaluations on downstream large VLMs, we use the OpenFlamingo-3B (OF-3B) (Awadalla et al., 2023), which aligned the CLIP image encoder (ViT-L from OpenAI) with the MPT-1B (Team et al., 2023), and LLaVA-7B (v1.5) (Liu et al., 2023a) which use the same image encoder as OF-3B, but aligned with the Vicuna-7B (Chiang et al., 2023). Additionally, we evaluate MiniGPT4-v2, which aligned the ViT-G-14 trained with EVA-CLIP (Fang et al., 2023) with Llama2 (Touvron et al., 2023) and BLIP2 use the same vision encoder and aligned with OPT (Zhang et al., 2022b). The summary of the large VLMs we used in the evaluations is summarised in Table 6. For large VLMs that use different image resolutions than our default $224 \times 224$, we use interpolation to rescale the perturbation to the resolution used by the VLM.

*Table 6.* Large Vision Language Models used in the experiments.

| Model Name | Image Encoder | LLM | Image Resolution |
|---|---|---|---|
| OpenFlamingo-3B (OF-3B) | ViT-L/14 CLIP OpenAI | MPT-1B | $224 \times 224$ |
| LLaVA-7B | ViT-L/14 CLIP OpenAI | Vicuna-7B | $224 \times 224$ |
| MiniGPT4-v2 | ViT-G/14 EVA-CLIP | Llama2 Chat 7B | $448 \times 448$ |
| BLIP2 | ViT-G/14 EVA-CLIP | OPT-6.7B | $364 \times 364$ |

**Baselines.** We compare our approach to state-of-the-art UAP methods tailored for CLIP encoders, including C-PGC[1] (Fang et al., 2024b), ETU[2] (Zhang et al., 2024), and AdvCLIP[3] (Zhou et al., 2023a). We also evaluate against UAPs originally designed for image classifiers, GD-UAP[4] (Mopuri et al., 2018), TRM-UAP[5] (Liu et al., 2023b), and Meta-UAP (Weng et al., 2024). All UAPs are either directly obtained from official open-source repositories or generated using the official code provided by each baseline's authors.

## C.2. Search Space

Our search space for CLIP surrogate encoders spans a wide range of architectures, pre-training datasets, objectives, and training recipes. For architectures, we include ResNet (RN) (He et al., 2016), ConvNext (Liu et al., 2022), ViT (Dosovitskiy et al., 2021), ViTamin (Chen et al., 2024b), NLLB (Visheratin, 2023), MobileCLIP (Vasu et al., 2024), and RoBERTa (Liu et al., 2019). For pre-training datasets, surrogate encoders are trained on diverse datasets, including CC12M (Changpinyo et al., 2021), YFCC15M (Thomee et al., 2016), LAION (Schuhmann et al., 2021; 2022), DataComp/CommonPool (Gadre et al., 2023), Merged2B (Sun et al., 2023), DFN (Fang et al., 2024a), WebLI (Zhai et al., 2023), and MetaCLIP (Xu et al.,

---

[1] https://github.com/ffhibnese/cpgc_vlp_universal_attacks
[2] https://github.com/sduzpf/UAP_VLP
[3] https://github.com/CGCL-codes/AdvCLIP
[4] https://github.com/val-iisc/GD-UAP
[5] https://github.com/MILO-GRP/TRM-UAP

*Table 7.* Details regarding each baseline method and different configurations.

| Method | Variant | Note |
|---|---|---|
| GD-UAP | Seg
CLS | Generated with segmentation model (ResNet152 backbone) as the surrogate.
Generated with ResNet152 classifier as the surrogate. |
| AdvCLIP | ViT-B/16
RN101 | Generated with CLIP ViT-B/16 released by OpenAI on the NUS-WIDE dataset.
Generated with CLIP ResNet101 released by OpenAI on the NUS-WIDE dataset. |
| TRM-UAP | GoogleNet
RN152 | Generated with GoogleNet classifier trained on ImageNet.
Generated with RN152 classifier trained on ImageNet. |
| Meta-UAP | Ensemble
Ensemble-Meta | Generated with an ensemble of DenseNet121, VGG16, and ResNet50 classifiers trained on ImageNet.
Same as above, with meta-learning strategy. |
| C-GPC | RN101-Flicker
RN101-COCO
ViT-B/16-Flicker
ViT-B/16-COCO | Generated with CLIP ResNet101 released by OpenAI on the Flicker dataset.
Generated with CLIP ResNet101 released by OpenAI on the MSCOCO dataset.
Generated with CLIP ViT-B/16 released by OpenAI on the Flicker dataset.
Generated with CLIP ViT-B/16 released by OpenAI on the MSCOCO dataset. |
| ETU | RN101-Flicker
ViT-B/16-Flicker | Generated with CLIP ResNet101 released by OpenAI on the Flicker dataset.
Generated with CLIP ViT-B/16 released by OpenAI on the Flicker dataset. |

2024). For pre-training objectives and recipes, we consider methods such as SigLIP (Zhai et al., 2023), EVA-CLIP (Sun et al., 2023), CLIPA (Li et al., 2023c), and COCA (Yu et al., 2022).

Details about the **Base**, **Mid**, and **Large** search spaces are provided in Tables 8, 9, and 10, respectively. The Base search space is balanced and comprises encoders from 4 diverse architecture types: ResNet (RN) (He et al., 2016), ConvNext (Liu et al., 2022), ViT-B, and ViT-L (Dosovitskiy et al., 2021), with 4 encoders per architecture. The Mid search space expands on the Base search space by incorporating additional ViT-L and ViT-B models available in OpenCLIP. The Large search space further includes larger models, such as ViT-H and ViT-bigG, augmenting the Mid search space. To ensure a strict black-box evaluation setting, there is no overlap between the surrogate encoders in these search spaces and the encoders used in evaluations (Tables 5 and 6).

*Table 8.* List of encoders in the Base search space, including each one's architecture, pre-training dataset, and the corresponding OpenCLIP identifier.

| | Architecture | Pre-training Dataset | OpenCLIP Identifier |
|---|---|---|---|
| 1 | RN101 | YFCC15M | (RN101, yfcc15m) |
| 2 | RN50 | YFCC15M | (RN50, yfcc15m) |
| 3 | RN50 | CC12M | (RN50, cc12m) |
| 4 | RN101-quickgelu | YFCC15M | (RN101-quickgelu, yfcc15m) |
| 5 | ConvNext Base | LAION400M | (convnext_base, laion400m_s13b_b51k) |
| 6 | ConvNext Base-W | LAION2B | (convnext_base_w, laion2b_s13b_b82k) |
| 7 | ConvNext Base-W | LAION2B | (convnext_base_w, laion2b_s13b_b82k_augreg) |
| 8 | ConvNext Large-D | LAION2B | (convnext_large_d, laion2b_s26b_b102k_augreg) |
| 9 | ViT-B/16 | DFN2B | (ViT-B-16, dfn2b) |
| 10 | ViT-B/16 | DataComp | (ViT-B-16, datacomp_xl_s13b_b90k) |
| 11 | ViT-B/32 | LAION2B | (ViT-B-32, laion2b_s34b_b79k) |
| 12 | ViT-B/32 | DataComp | (ViT-B-32, datacomp_xl_s13b_b90k) |
| 13 | ViT-L/14 | LAION400M | (ViT-L-14, laion400m_e32) |
| 14 | ViT-L/14 | CommonPool | (ViT-L-14, commonpool_xl_s13b_b90k) |
| 15 | EVA02-L/14 | Merged2B | (EVA02-L-14, merged2b_s4b_b131k) |
| 16 | ViT-L/14-CLIPA | DataComp | (ViT-L-14-CLIPA, datacomp1b) |

*Table 9.* List of encoders in the Mid search space, including each one's architecture, pre-training dataset, and the corresponding OpenCLIP identifier.

| | Architecture | Pre-training Dataset | OpenCLIP Identifier |
|---|---|---|---|
| 1 | RN101 | YFCC15M | (RN101, yfcc15m) |
| 2 | RN50 | YFCC15M | (RN50, yfcc15m) |
| 3 | RN50 | CC12M | (RN50, cc12m) |
| 4 | ConvNext Base | LAION400M | (convnext_base, laion400m_s13b_b51k) |
| 5 | ConvNext Base-W | LAION2B | (convnext_base_w, laion2b_s13b_b82k) |
| 6 | ConvNext Base-W | LAION2B | (convnext_base_w, laion2b_s13b_b82k_augreg) |
| 7 | ConvNext Base-W | LAION Aesthetic | (convnext_base_w, laion_aesthetic_s13b_b82k) |
| 8 | ConvNext Large-D | LAION2B | (convnext_large_d, laion2b_s26b_b102k_augreg) |
| 9 | ConvNext XXLarge | LAION2B | (convnext_xxlarge, laion2b_s34b_b82k_augreg) |
| 10 | ViT-B/32 | LAION400M | (ViT-B-32, laion400m_e31) |
| 11 | ViT-B/32 | LAION400M | (ViT-B-32, laion400m_e32) |
| 12 | ViT-B/32 | LAION2B | (ViT-B-32, laion2b_e16) |
| 13 | ViT-B/32 | LAION2B | (ViT-B-32, laion2b_s34b_b79k) |
| 14 | ViT-B/32 | DataComp | (ViT-B-32, datacomp_xl_s13b_b90k) |
| 15 | ViT-B/16 | LAION400M | (ViT-B-16, laion400m_e31) |
| 16 | ViT-B/16 | LAION400M | (ViT-B-16, laion400m_e32) |
| 17 | ViT-B/16 | LAION2B | (ViT-B-16, laion2b_s34b_b88k) |
| 18 | ViT-B/16 | DataComp | (ViT-B-16, datacomp_xl_s13b_b90k) |
| 19 | ViT-B/16 | DataComp | (ViT-B-16, datacomp_l_s1b_b8k) |
| 20 | ViT-B/16 | DFN2B | (ViT-B-16, dfn2b) |
| 21 | EVA02-B/16 | Merged2B | (EVA02-B-16, merged2b_s8b_b131k) |
| 22 | ViT-L/14 | LAION400M | (ViT-L-14, laion400m_e31) |
| 23 | ViT-L/14 | LAION400M | (ViT-L-14, laion400m_e32) |
| 24 | ViT-L/14 | LAION2B | (ViT-L-14, laion2b_s32b_b82k) |
| 25 | ViT-L/14 | DataComp | (ViT-L-14, datacomp_xl_s13b_b90k) |
| 26 | ViT-L/14 | CommonPool | (ViT-L-14, commonpool_xl_clip_s13b_b90k) |
| 27 | ViT-L/14 | CommonPool | (ViT-L-14, commonpool_xl_laion_s13b_b90k) |
| 28 | ViT-L/14 | CommonPool | (ViT-L-14, commonpool_xl_s13b_b90k) |
| 29 | ViT-L/14 | DFN2B | (ViT-L-14, dfn2b) |
| 30 | EVA02-L-14 | Merged2B | (EVA02-L-14, merged2b_s4b_b131k) |
| 31 | ViT-SO400M-14-SigLIP | WebLI | (ViT-SO400M-14-SigLIP, webli) |
| 32 | ViT-L/14-CLIPA | DataComp | (ViT-L-14-CLIPA, datacomp1b) |

*Table 10.* List of encoders in the Large search space, including each one's architecture, pre-training dataset, and the corresponding OpenCLIP identifier.

| | Architecture | Pre-training Dataset | OpenCLIP Identifier |
|---|---|---|---|
| 1 | RN101 | YFCC15M | (RN101, yfcc15m) |
| 2 | RN50 | YFCC15M | (RN50, yfcc15m) |
| 3 | RN50 | CC12M | (RN50, cc12m) |
| 4 | ConvNext Base | LAION400M | (convnext_base, laion400m_s13b_b51k) |
| 5 | ConvNext Base-W | LAION2B | (convnext_base_w, laion2b_s13b_b82k) |
| 6 | ConvNext Base-W | LAION2B | (convnext_base_w, laion2b_s13b_b82k_augreg) |
| 7 | ConvNext Base-W | LAION Aesthetic | (convnext_base_w, laion_aesthetic_s13b_b82k |
| 8 | ConvNext Large-D | LAION2B | (convnext_large_d, laion2b_s26b_b102k_augreg) |
| 9 | ConvNext XXLarge | LAION2B | (convnext_xxlarge, laion2b_s34b_b82k_augreg) |
| 10 | ConvNext XXLarge | LAION2B | (convnext_xxlarge, laion2b_s34b_b82k_augreg_rewind) |
| 11 | ConvNext XXLarge | LAION2B | (convnext_xxlarge, laion2b_s34b_b82k_augreg_soup) |
| 12 | ViT-B/32 | LAION400M | (ViT-B-32, laion400m_e31) |
| 13 | ViT-B/32 | LAION400M | (ViT-B-32, laion400m_e32) |
| 14 | ViT-B/32 | LAION2B | (ViT-B-32, laion2b_e16) |
| 15 | ViT-B/32 | LAION2B | (ViT-B-32, laion2b_s34b_b79k) |
| 16 | ViT-B/32 | DataComp | (ViT-B-32, datacomp_xl_s13b_b90k) |
| 17 | ViT-B/32 | MetaCLIP | (ViT-B-32-quickgelu, metaclip_400m) |
| 18 | ViT-B/32 | MetaCLIP | (ViT-B-32-quickgelu, metaclip_fullcc) |
| 19 | ViT-B/16 | LAION400M | (ViT-B-16, laion400m_e31) |
| 20 | ViT-B/16 | LAION400M | (ViT-B-16, laion400m_e32) |
| 21 | ViT-B/16 | LAION2B | (ViT-B-16, laion2b_s34b_b88k) |
| 22 | ViT-B/16 | DataComp | (ViT-B-16, datacomp_xl_s13b_b90k) |
| 23 | ViT-B/16 | DataComp | (ViT-B-16, datacomp_l_s1b_b8k) |
| 24 | ViT-B/16 | CommonPool | (ViT-B-16, commonpool_l_s1b_b8k) |
| 25 | ViT-B/16 | DFN2B | (ViT-B-16, dfn2b) |
| 26 | ViT-B/16 | MetaCLIP | (ViT-B-16-quickgelu, metaclip_400m) |
| 27 | ViT-B/16 | MetaCLIP | (ViT-B-16-quickgelu, metaclip_fullcc) |
| 28 | EVA02-B/16 | Merged2B | (EVA02-B-16, merged2b_s8b_b131k) |
| 29 | ViT-L/14 | LAION400M | (ViT-L-14, laion400m_e31) |
| 30 | ViT-L/14 | LAION400M | (ViT-L-14, laion400m_e32) |
| 31 | ViT-L/14 | LAION2B | (ViT-L-14, laion2b_s32b_b82k) |
| 32 | ViT-L/14 | DataComp | (ViT-L-14, datacomp_xl_s13b_b90k) |
| 33 | ViT-L/14 | CommonPool | (ViT-L-14, commonpool_xl_clip_s13b_b90k) |
| 34 | ViT-L/14 | CommonPool | (ViT-L-14, commonpool_xl_laion_s13b_b90k) |
| 35 | ViT-L/14 | CommonPool | (ViT-L-14, commonpool_xl_s13b_b90k) |
| 36 | ViT-L/14 | MetaCLIP | (ViT-L-14-quickgelu, metaclip_400m) |
| 37 | ViT-L/14 | MetaCLIP | (ViT-L-14-quickgelu, metaclip_fullcc) |
| 38 | ViT-L/14 | DFN2B | (ViT-L-14, dfn2b) |
| 39 | EVA02-L-14 | Merged2B | (EVA02-L-14, merged2b_s4b_b131k) |
| 40 | ViT-SO400M-14-SigLIP | WebLI | (ViT-SO400M-14-SigLIP, webli) |
| 41 | ViT-L/14-CLIPA | DataComp | (ViT-L-14-CLIPA, datacomp1b) |
| 42 | ViT-H/14 | LAION2B | (ViT-H-14, laion2b_s32b_b79k) |
| 43 | ViT-H/14 | MetaCLIP | (ViT-H-14-quickgelu, metaclip_fullcc) |
| 44 | ViT-H/14 | DFN5B | (ViT-H-14-quickgelu, dfn5b) |
| 45 | ViT-G/14 | LAION2B | (ViT-g-14, laion2b_s12b_b42k) |
| 46 | ViT-G/14 | LAION2B | (ViT-g-14, laion2b_s34b_b88k) |
| 47 | EVA01-G/14-plus | Merged2B | (EVA01-g-14-plus, merged2b_s11b_b114k) |
| 48 | EVA02-E-14-plus | LAION2B | (EVA02-E-14-plus, laion2b_s9b_b144k) |
| 49 | ViT-bigG/14 | LAION2B | (ViT-bigG-14, laion2b_s39b_b160k) |
| 50 | ViT-bigG-14-CLIPA | DataComp | (ViT-bigG-14-CLIPA, datacomp1b) |
| 51 | ROBERTA-ViT-B/32 | LAION2B | (roberta-ViT-B-32, laion2b_s12b_b32k) |
| 52 | XLM-ROBERTA-VIT-B/32 | LAION5B | (xlm-roberta-base-ViT-B-32, laion5b_s13b_b90k) |
| 53 | XLM-ROBERTA-Large-VIT-H/14 | LAION5B | (xlm-roberta-large-ViT-H-14, frozen_laion5b_s13b_b90k) |
| 54 | NLLB-Base | NLLB | (nllb-clip-base, v1) |
| 55 | NLLB-Large | NLLB | (nllb-clip-large, v1) |
| 56 | ViTamin-B | DataComp | (ViTamin-B, datacomp1b) |
| 57 | ViTamin-B-LTT | DataComp | (ViTamin-B-LTT, datacomp1b) |
| 58 | ViTamin-L | DataComp | (ViTamin-L, datacomp1b) |
| 59 | ViTamin-L2 | DataComp | (ViTamin-L2, datacomp1b) |
| 60 | MobileCLIP-S1 | DataComp | (MobileCLIP-S1, datacompdr) |
| 61 | MobileCLIP-S2 | DataComp | (MobileCLIP-S2, datacompdr) |
| 62 | MobileCLIP-B | DataComp | (MobileCLIP-B, datacompdr) |
| 63 | COCA ViT-L/14 | LAION2B | (coca_ViT-L-14, laion2b_s13b_b90k) |
| 64 | COCA ViT-B/32 | LAION2B | (coca_ViT-B-32, laion2b_s13b_b90k) |

## C.3. Extended Results

In Figures 5, 6, and 7, we present a detailed comparison of X-Transfer against baseline methods across 9 victim CLIP encoders. For baselines such as ETU (Zhang et al., 2024) and C-PGC (Fang et al., 2024b), which use ViT-B/16 trained by OpenAI (Radford et al., 2021) as the surrogate model, we exclude results for ViT-B/16 as the victim model, as this configuration constitutes a white-box setting. Each baseline's results reflect its best performance across all tested configurations. Notably, results shows that X-Transfer achieves state-of-the-art performance on all 9 victim encoders for both zero-shot classification and image-text retrieval tasks.

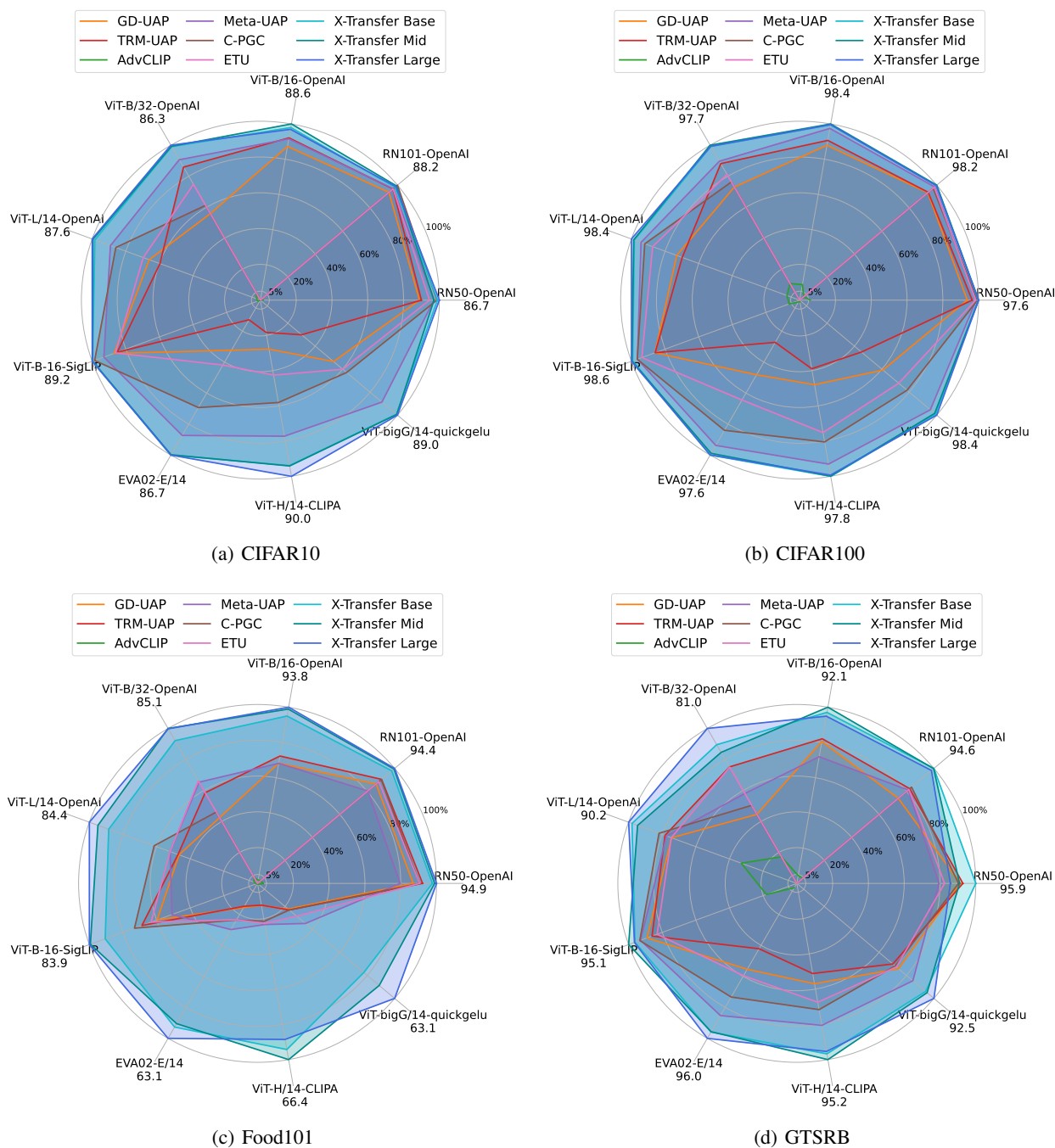

*Figure 5.* Detailed results on the non-targeted attack success rate for 9 victim encoders in a zero-shot classification task, evaluated on the following datasets: (a) CIFAR-10, (b) CIFAR-100, (c) Food101, and (d) GTSRB.

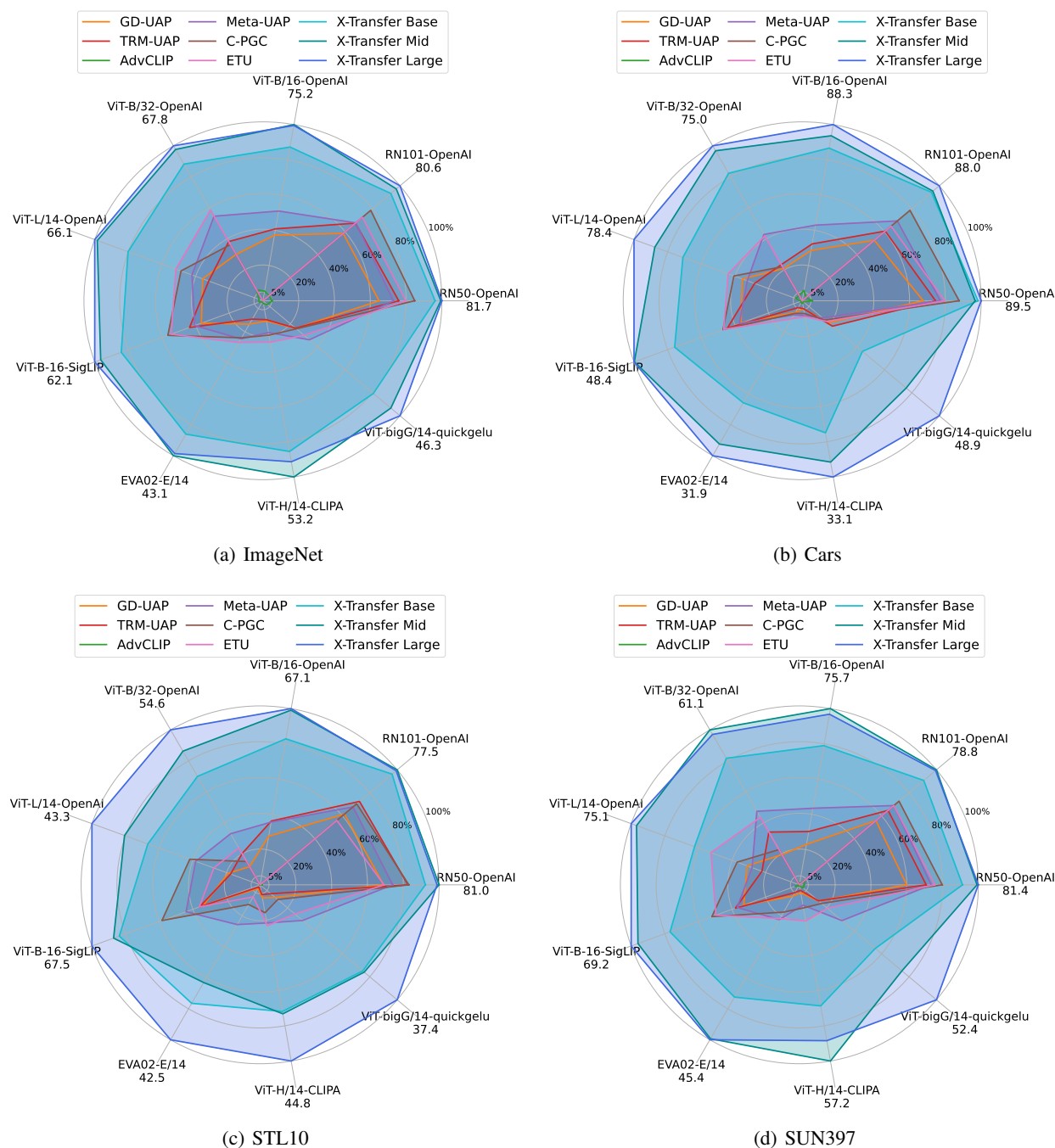

*Figure 6.* Detailed results on the non-targeted attack success rate for 9 victim encoders in a zero-shot classification task, evaluated on the following datasets: (a) ImageNet, (b) Stanford Cars, (c) STL10, and (d) SUN397.

## C.4. Scaling with ETU

In Table 11, we present the results of scaling with our adversarial objective in conjunction with ETU (Zhang et al., 2024), which utilises a specialised loss function designed for CLIP encoders. We denote the application of our efficient search algorithm with the ETU loss function as 'ETU+X-Transfer' using the Base search space. For comparison, we include X-Transfer Vanilla, which does not use scaling and solely employs our adversarial objective, as well as X-Transfer Base. The results indicate that our generic adversarial objective function is essential for effectively scaling up the number of surrogate models, enabling super transferability across diverse settings.

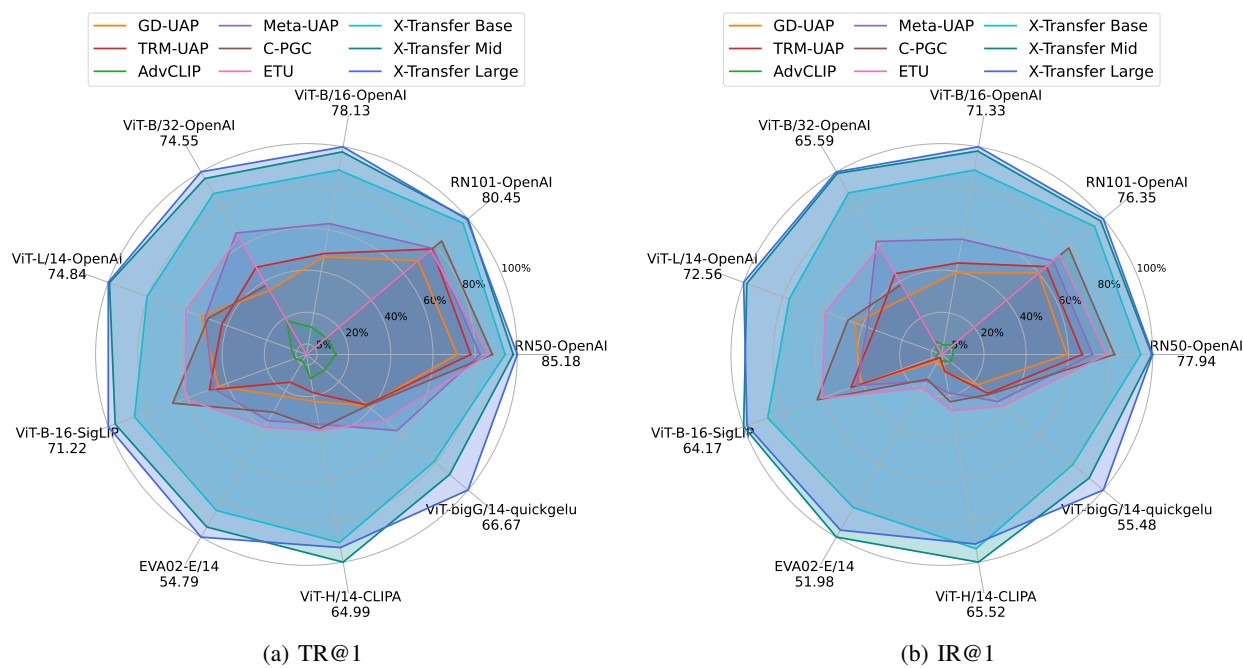

(a) TR@1          (b) IR@1

*Figure 7.* Detailed results on the non-targeted attack success rate for 9 victim encoders in the image-text retrieval task, evaluated on the MSCOCO, with metric (a) TR@1 and (b) IR@1.

*Table 11.* Comparison between scaling with ETU and scaling with our adversarial objective function.

| Method | Search Space | Zero-Shot Classification | | | | | | | | | I-T Retrieval | |
|---|---|---|---|---|---|---|---|---|---|---|---|---|
| | | C-10 | C-100 | Food | GTSRB | ImageNet | Cars | STL | SUN | Avg | TR@1 | IR@1 |
| ETU (Zhang et al., 2024) | None ($N = 1$) | 70.2 | 86.5 | 47.1 | **71.1** | **34.1** | **31.1** | **27.5** | **31.0** | **49.8** | 40.2 | **32.8** |
| ETU + X-Transfer | Base ($N = 16$) | **73.6** | **88.6** | **48.8** | 69.0 | 31.9 | 27.2 | 21.7 | 24.7 | 48.2 | **41.6** | 32.1 |
| X-Transfer Vanilla | None ($N = 1$) | 72.7 | 88.3 | 49.9 | 72.3 | 31.2 | 26.3 | 19.2 | 27.6 | 48.4 | 42.3 | 34.5 |
| X-Transfer Base | Base ($N = 16$) | **86.6** | **97.5** | **74.8** | **89.3** | **56.0** | **52.1** | **46.8** | **50.7** | **69.2** | **63.7** | **58.8** |

## C.5. Comparing Surrogate Dataset

In Table 12, we present the results of using MSCOCO (Lin et al., 2014) as a surrogate dataset and compare them with ImageNet. The results show that the ASR is comparably similar across both datasets. This finding indicates that super transferability does not depend on the choice of the surrogate dataset. Furthermore, in Appendix C.9, we demonstrate that the patterns generated in the UAP are also independent of the surrogate dataset. Instead, it is the surrogate encoders that primarily influence both the transferability and the patterns in the UAP.

*Table 12.* Comparison between surrogate datasets used for generating UAPs using X-Transfer.

| Surrogate Dataset | Zero-Shot Classification | | | | | | | | | I-T Retrieval | |
|---|---|---|---|---|---|---|---|---|---|---|---|
| | C-10 | C-100 | Food | GTSRB | ImageNet | Cars | STL | SUN | Avg | TR@1 | IR@1 |
| ImageNet | 86.6 | 97.5 | 74.8 | 89.3 | 56.0 | 52.1 | 46.8 | 50.7 | 69.2 | 63.7 | 58.5 |
| MSCOCO | 84.9 | 96.2 | 69.7 | 82.4 | 53.1 | 49.7 | 37.3 | 49.9 | 65.4 | 61.9 | 57.8 |

## C.6. UAP with $L_2$-norm Perturbations and Adversarial Patch

In table 13, we present the results for using $L_2$-norm perturbation and adversarial patch introduced in Appendix B. The results show consistent conclusions that the efficient scaling of X-Transfer can improve super transferability.

Table 13. The non-targeted ASR (%) results for $L_2$-norm perturbation and adversarial patch.

| Type | Search Space | Zero-Shot Classification | | | | | | | | | I-T Retrieval | |
|---|---|---|---|---|---|---|---|---|---|---|---|---|
| | | C-10 | C-100 | Food | GTSRB | ImageNet | Cars | STL | SUN | Avg | TR@1 | IR@1 |
| $L_2$ | Base ($N = 16$) | 85.7 | 97.5 | 77.5 | 90.1 | 64.6 | 61.0 | 50.3 | 65.5 | 74.0 | 76.0 | 69.9 |
| | Mid ($N = 32$) | 86.8 | **98.4** | 86.7 | 90.1 | 83.7 | 73.1 | 60.7 | 84.0 | 82.9 | 88.6 | 81.3 |
| | Large ($N = 64$) | **88.2** | **98.4** | **95.5** | **92.1** | **88.5** | **83.9** | **81.1** | **90.2** | **89.7** | **93.6** | **88.5** |
| Patch | Base ($N = 16$) | 51.5 | 83.9 | 39.5 | 75.9 | 48.1 | 46.7 | 10.9 | 42.9 | 49.9 | 64.2 | 49.0 |
| | Mid ($N = 32$) | 55.7 | 85.1 | 51.0 | 80.1 | 54.9 | 48.8 | 15.3 | 56.1 | 55.9 | 65.7 | 55.0 |
| | Large ($N = 64$) | **88.6** | **98.5** | **98.9** | **93.7** | **99.9** | **99.3** | **89.7** | **99.6** | **96.0** | **100.0** | **99.9** |

## C.7. TUAP

In this section, we present the evaluation results of the X-Transfer attack with a targeted objective, where the adversary can specify any text description as the target.

**Target Text Description.** We use a total of 10 target text descriptions for evaluating TUAP. Targets No.1 to No.6 are adopted from existing works (Schlarmann & Hein, 2023; Schlarmann et al., 2024). We constructed the rest of the targets ourselves.

**Evaluations.** We adopt the standard zero-shot classification setup and utilise the template provided by Radford et al. (2021) for each evaluation dataset. For instance, we use the format "an image of $X$," where $X$ is replaced by the class name. To evaluate TUAP, we measure the attack success rate. For each dataset, we include an additional class representing the adversary's target and replace $X$ with the target text description. TUAP is applied to each image in the evaluation dataset, and the victim model generates image embeddings. The attack is deemed successful if the closest embedding matches the template containing the adversarial target text description.

For image retrieval, we randomly select an image, apply perturbation, and use an adversary-specified target text sentence as the query. We report the rank of the perturbed image among all images as the Image Retrieval Rank (IR Rank), where a lower rank signifies a more successful TUAP. For the MSCOCO dataset, which contains 3,900 images, we repeat the retrieval process 50 times for each attack type, victim model, and target text sentence. The results are reported as the mean and standard deviation of the IR Rank.

For image captioning and VQA tasks evaluated with large VLMs, we use the widely adopted CIDEr metric (Vedantam et al., 2015) for captioning tasks and VQA accuracy for question answering. Additionally, for image captioning, we report the BLEU-4 score (Papineni et al., 2002) to measure the similarity between the generated caption and the adversary's target text description as the targeted ASR. A higher BLEU-4 score indicates greater alignment with the target text. BLEU-4 scores are omitted for VQA tasks due to the brevity of the answers. In both tasks, we apply the TUAP to each image in the evaluation dataset and assess the victim VLM's response.

**Results.** In Table 14, we present the results of our TUAP on the zero-shot classification and image-text retrieval tasks. The results demonstrate that the scaling capability aligns with our analysis in the main paper for the non-targeted objective. Notably, in this case, the attack is deemed successful only if the prediction matches the text description specified by the adversary, which is inherently more challenging than a non-targeted objective that merely causes arbitrary errors. In Table 15, we provide the results of evaluating image captioning and VQA tasks on large VLMs. These results similarly exhibit consistent scaling capabilities. Furthermore, the responses generated by the large VLMs closely match the targeted text description, as measured by the targeted ASR (BLEU-4 score between the response and the target text description). A qualitative example is provided in Section 4.3. These findings indicate that X-Transfer is capable of generating both non-targeted UAPs, which cause arbitrary errors, and targeted UAPs, which align predictions with adversary-specified text descriptions.

**Relation with related works.** The TUAP threat model presented in this section is related to concurrent works (Shayegani et al., 2023; Carlini et al., 2023; Wang et al., 2024b; Tao et al., 2024; Wang et al., 2024c; Gong et al., 2025) that explore jailbreak attacks against large VLMs, which typically involve a collection of harmful or adversarial "targets." However, our threat model differs fundamentally, and we target the CLIP encoder directly rather than downstream VLMs. These differences in focus and attack surface make direct comparisons with existing jailbreak methods unfair. Nevertheless, we believe that TUAPs generated by X-Transfer on CLIP, especially when aligned with jailbreak-style prompts, could serve as

*Table 14.* The targeted ASR (%) results in zero-shot classification and image-text (I-T) retrieval tasks across different CLIP encoders and datasets. I-T retrieval is evaluated on MSCOCO. Results are based on averaging over 9 black-box victim encoders and 10 target text descriptions.

| Search Space | Zero-Shot Classification | | | | | | | | | I-T Retrieval | |
|---|---|---|---|---|---|---|---|---|---|---|---|
| | C-10 | C-100 | Food | GTSRB | ImageNet | Cars | STL | SUN | Avg | TR@1 | IR Rank |
| Base ($N=16$) | 99.9 | 99.1 | 72.0 | 98.0 | 50.1 | 44.2 | 89.6 | 53.7 | 75.8 | 35.3 | 233.0 |
| Mid ($N=32$) | **100.0** | 99.6 | 77.9 | 98.3 | 56.0 | 54.4 | 90.9 | 60.1 | 79.7 | **42.9** | 206.0 |
| Large ($N=64$) | **100.0** | **99.8** | **79.6** | **98.9** | **57.2** | **54.5** | **92.7** | **60.4** | **80.4** | 42.3 | **167.5** |

*Table 15.* Non-targeted ASR (%) and BLEU-4 results in image captioning and VQA across various large VLMs and datasets. For image captioning, CIDEr is used as the evaluation metric, while VQA accuracy is employed for the VQA task. Results are based on $L_\infty$-norm bounded perturbations and are averaged across 10 different target descriptions.

| Method | Victim Model | COCO | | Flickr-30K | | OK-VQA | WizViz |
|---|---|---|---|---|---|---|---|
| | | Non-targeted ASR | BLEU-4 | Non-targeted ASR | BLEU-4 | Non-targeted ASR | |
| Base | | 61.0±5.0 | 13.6±6.1 | 56.4±3.5 | 12.0±5.7 | 36.3±3.2 | 37.4±2.2 |
| Mid | OF-3B | 68.0±4.5 | 16.4±7.1 | 63.9±3.8 | 14.0±6.6 | 40.7±5.1 | 41.3±5.0 |
| Large | | **70.7±4.7** | **17.8±9.0** | **66.0±4.4** | **15.3±7.4** | **41.0±6.7** | **41.8±5.7** |
| Base | | 39.5±6.9 | 12.2±6.5 | 35.1±6.3 | 11.8±6.7 | 16.8±4.1 | **20.2±6.5** |
| Mid | LLaVA-7B | 41.5±5.0 | 13.0±7.0 | 37.0±4.6 | 12.5±7.1 | 17.4±3.0 | 19.7±4.6 |
| Large | | **45.9±3.4** | **14.2±7.6** | **40.3±3.0** | **13.1±7.1** | **18.9±3.1** | 15.4±7.8 |
| Base | | 34.2±5.7 | 9.6±6.5 | 30.7±3.9 | 10.0±7.0 | 17.4±2.8 | 11.0±3.8 |
| Mid | MiniGPT4 | 35.3±3.7 | 9.9±6.4 | 31.0±2.8 | 10.2±6.9 | 17.5±2.3 | 9.7±4.8 |
| Large | | **40.2±3.6** | **11.3±7.0** | **35.1±2.7** | **11.2±7.1** | **18.7±2.8** | **11.6±6.8** |
| Base | | 50.5±7.1 | 13.4±6.7 | 43.2±6.5 | 11.9±6.8 | 48.9±6.2 | 64.6±2.8 |
| Mid | BLIP2 | **53.0±7.0** | **15.7±8.6** | **46.0±6.8** | **13.8±8.4** | **50.6±7.0** | **64.8±4.9** |
| Large | | 52.2±6.5 | 15.2±9.3 | 45.4±6.6 | 13.8±9.1 | 45.0±8.0 | 62.5±6.8 |

effective initialisation points for future jailbreak attacks targeting large VLMs.

In parallel, TUAP can also be viewed as a form of backdoor trigger (Carlini & Terzis, 2022; Jia et al., 2022; Liang et al., 2024; Bai et al., 2024; Huang et al., 2025). However, unlike these attacks, TUAP achieves its objective without data poisoning or any training-time manipulation. Instead, it operates as a test-time backdoor trigger (Lu et al., 2024), making it a novel and distinct type of safety vulnerability in pre-trained vision encoders. This shift from training-time to test-time attack surfaces introduces new challenges securing multi-modal models.

## C.8. Evaluation on Adversarial Trained Encoders

In this section, we analyse UAPs generated by X-Transfer when evaluated with adversarially trained CLIP models. Mao et al. (2023) proposed a supervised adversarial training method fine-tuned on ImageNet, and its performance was further improved through unsupervised fine-tuning (Schlarmann et al., 2024). For our evaluations, we include two adversarially trained CLIP image encoders: FARE-2 (Schlarmann et al., 2024) and TeCoA-2 (Mao et al., 2023). The suffix "-2" indicates that the models were trained with $L_\infty$-norm perturbations bounded to $\frac{2}{255}$.

As shown in Table 16, adversarial training can defend against $L_\infty$-norm bounded UAPs, which aligns with existing literature. It is well established that adversarial training can defend against universal perturbations (Weng et al., 2024) and provides resistance to black-box attacks. However, our results demonstrate that these adversarially trained models remain vulnerable to adversarial patches and $L_2$-norm perturbations. This is likely because they are specifically trained to counter only $L_\infty$-norm bounded attacks. The adversarial training is robust to multiple different types of perturbations (Kang et al., 2019; Croce & Hein, 2020a; Hsiung et al., 2023), which have not been explored in CLIP.

These findings suggest that while adversarial training can partially mitigate CLIP's vulnerabilities, it does not provide comprehensive robustness. Furthermore, adversarial training often requires a trade-off between clean zero-shot accuracy and robustness (Tsipras et al., 2019) and is known for being computationally expensive, limiting its scalability on web-scale

datasets.

Table 16. The non-targeted ASR (%) results for the evaluation of CLIP encoders are fine-tuned with adversarial training.

| Type | Victim Encoder | Zero-Shot Classification | | | | | | | | |
|------|------|------|------|------|------|------|------|------|------|------|
| | | C-10 | C-100 | Food | GTSRB | ImageNet | Cars | STL | SUN | Avg |
| $L_\infty$ | FARE | 14.2 | 23.7 | 17.9 | 33.0 | 5.9 | 4.5 | 1.9 | 6.4 | 13.4 |
| | TeCoA | 3.7 | 5.7 | 12.6 | 17.9 | 3.0 | 4.2 | 1.2 | 4.8 | 6.6 |
| $L_2$ | FARE | 76.1 | 93.1 | 64.8 | 71.0 | 47.4 | 34.3 | 23.2 | 56.3 | 58.3 |
| | TeCoA | 27.0 | 46.0 | 34.6 | 46.4 | 13.9 | 19.5 | 6.9 | 21.6 | 27.0 |
| Patch | FARE | 88.8 | 98.2 | 98.8 | 95.0 | 99.9 | 99.3 | 89.8 | 99.8 | 96.2 |
| | TeCoA | 88.5 | 98.2 | 98.2 | 94.1 | 99.9 | 98.9 | 89.6 | 99.2 | 95.8 |

## C.9. Qualitative Analysis on UAPs

The visual clarity of the UAP patterns scales with the search space size $N$ in the X-Transfer attack, as shown in Figure 8(a). This further supports the observation that visual clarity correlates with the attack success rate. Additional visualisations of the effects of UAPs and TUAPs applied to images, along with the corresponding responses from VLMs, are provided in Figure 9. The visualisations of TUAPs are presented in Figure 10.

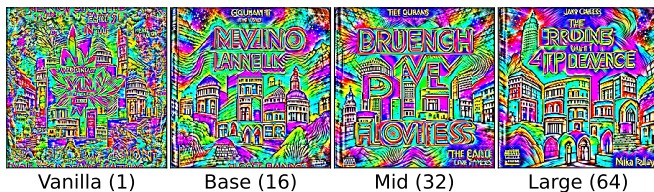

Vanilla (1)   Base (16)   Mid (32)   Large (64)

(a) X-transfer with different search space ($N$).

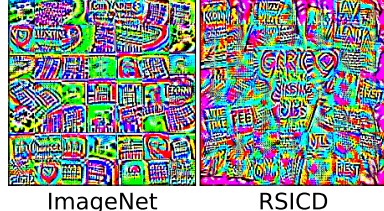

ImageNet   RSICD

(b) X-transfer with encoders trained on RSICD dataset, with ImageNet (left) or RSICD (right) as surrogate dataset.

Figure 8. The visualisation of UAPs in (a) scaling with $N$ of X-Transfer, and (b) generated with encoders trained on RSICD dataset. All UAPs are generated with the Base search space.

Table 17. The predicted class with UAP generated by X-Transfer is applied to images in each dataset. Results are based on CLIP encoder ViT-L/14 released by OpenAI.

| Dataset | Top Predicted Class |
|------|------|
| CIFAR-10 | Ship (87.6%), Airplane (8.6%) |
| CIFAR-100 | Ray (99.2%) |
| Food101 | Cheese plate (16.7%), Grilled cheese sandwich (14.6%), Cheesecake (12.4%) |
| GTSRB | Red and white triangle with snowflake / ice warning (76.0%), Stop (4.3%) |
| ImageNet | Jay (18.0%), Dust jacket (3.4%), Kuvasz (2.9%) |
| Cars | Bentley Arnage Sedan 2009 (9.8%), Honda Odyssey Minivan 2012 (8.0%) |
| SUN397 | Cheese factory (30.7%), Trench (10.7%), Discotheque (8.8%) |

In this subsection, we further explore the origin of the UAP patterns generated by X-Transfer. We hypothesise that most UAPs generated exhibit building-like patterns due to the pre-training datasets of the surrogate encoders in our search space. To investigate this, we conducted experiments using CLIP encoders fine-tuned on the Remote Sensing Image Captioning Dataset (RSICD) (Lu et al., 2017). Using 4 encoders[6] trained on remote sensing imagery as the search space, we applied X-Transfer to generate UAPs. The resulting perturbations, shown in Figure 8(b), exhibit patterns resembling remote sensing imagery. This supports our hypothesis that the semantic characteristics of UAPs are influenced by the pre-training datasets of the surrogate encoders.

To explore further, we applied UAPs generated by X-Transfer to various datasets and used CLIP encoders for zero-shot classification. Surprisingly, while the UAP patterns visually resemble building-like structures, their predicted classes often lack any connection to buildings. As detailed in Table 17, CLIP encoders frequently predict concepts that seem unrelated to human interpretations of the perturbation. For example, on the SUN397 and Food101 datasets, predictions skew toward cheese-related concepts, while on the GTSRB dataset, predictions often correspond to the "ice warning" traffic sign, possibly due to semantic similarities to cheese-like textures. Unlike targeted attacks, these UAPs do not consistently steer the encoder toward a specific class. On datasets like CIFAR and ImageNet, the predictions vary significantly.

These findings indicate a distinction between the apparent semantic content of UAP patterns and their adversarial impact on CLIP encoders. While humans interpret these patterns as meaningful (e.g., building-like), they mislead CLIP encoders into producing diverse and often uninterpretable predictions.

As for why building-like patterns are predominant, we posit that such patterns are inherently more adversarial for CLIP encoders. The surrogate encoders used in our Base, Mid, and Large search spaces were predominantly pre-trained on web-scale datasets sourced from platforms like Common Crawl[7]. However, datasets like LAION (Schuhmann et al., 2021; 2022) and DataComp (Gadre et al., 2023) lack detailed disclosures of their image distributions, and their massive scale makes comprehensive analysis difficult. We speculate that building-like patterns are particularly adversarial for deceiving CLIP encoders trained on such datasets. We believe these findings open a promising avenue for future research, particularly in understanding the connection between pre-training data distributions and adversarial patterns.

---

[6]https://huggingface.co/flax-community/clip-rsicd
[7]https://commoncrawl.org

**Non-targeted**

**Clean**

**OF-3B:** A view of a lake in the middle of a mountain range.
**LLaVA-7B:** A river with mountains in the background.
**MiniGPT4:** A river running through a grassy field.
**BLIP2:** The sun shines over a stream in the mountains.

**Adversarial**

**OF-3B:** The Ruining of Avenge by Nikki Foley is licensed under a Creative Commons Attribution.
**LLaVA-7B:** A book cover with a picture of a river and a building.
**MiniGPT4:** The cover of the book the river of dreams.
**BLIP2:** The cover of the book, the riding to the stars.

**Targeted:** A great white shark flying over a bridge.

**Clean**

**OF-3B:** A view of downtown Vancouver, British Columbia, Canada.
**LLaVA-7B:** A cityscape at night with a large body of water in the background.
**MiniGPT4:** A city skyline at dusk with a river in the background.
**BLIP2:** An aerial view of a city at dusk.

**Adversarial**

**OF-3B:** A large shark swims in the ocean in front of a skyscraper.
**LLaVA-7B:** A shark is flying over a city.
**MiniGPT4:** A city skyline with a shark in the sky.
**BLIP2:** A photo of a shark flying over a city.

**Non-targeted**

**Clean**

**OF-3B:** A fire falls from the sky in Yosemite National Park, California.
**LLaVA-7B:** A waterfall with a yellow glow in the middle of a mountain.
**MiniGPT4:** A waterfall is flowing down a rocky cliff.
**BLIP2:** The sun shines over a stream in the mountains.

**Adversarial**

**OF-3B:** Nika Rally - The Trouble With 4P LEVELEN (2017) [FL
**LLaVA-7B:** A book cover with a purple sky and a building with a light. shining out of it.
**MiniGPT4:** A book cover with a city scene and the words the freedom of. aptitude.
**BLIP2:** The cover art for the album by nika palley.

**Targeted:** The Godzilla is swimming in the harbour.

**Clean**

**OF-3B:** Aurora Borealis, also known as the Northern Lights, is a natural light show.
**LLaVA-7B:** A house is surrounded by snow and trees.
**MiniGPT4:** The northern lights are seen in the sky above a house.
**BLIP2:** Aurora borealis in Norway.

**Adversarial**

**OF-3B:** A giant Godzilla is seen in the background.
**LLaVA-7B:** A picture of a monster with a green light behind it.
**MiniGPT4:** A giant monster is in the water with a city in the background.
**BLIP2:** Godzilla za wallpaper.

**Non-targeted**

**Clean**

**OF-3B:** The sun sets over the Summer Palace in Beijing, China.
**LLaVA-7B:** A beautiful sunset over a lake with a reflection of a building.
**MiniGPT4:** A sunset view of a building with a lake in the foreground.
**BLIP2:** The forbidden city at sunset in China.

**Adversarial**

**OF-3B:** The Ruinine with 4Tp DeVanee.
**LLaVA-7B:** The cover of the album The Ridin' with 4 TP DeVance.
**MiniGPT4:** A painting of a city with the words burning of applenpiece.
**BLIP2:** The cover art for the album, the riffs with a 4th degree.

**Targeted:** I am going to destroy all of your belongings.

**Clean**

**OF-3B:** A view of the city of London from the Shard, the tallest building in Western Europe.
**LLaVA-7B:** A city skyline with a river running through it.
**MiniGPT4:** A view of the city from a high vantage point.
**BLIP2:** An aerial view of London at night.

**Adversarial**

**OF-3B:** I am all of those things.
**LLaVA-7B:** A cityscape with a bunch of luggage and bags.
**MiniGPT4:** A city skyline with a lot of buildings and a sign that says i am a petroleum.
**BLIP2:** I am of destruction.

*Figure 9.* An illustration showing the application of both UAP (left) and TUAP (right) to an image. The responses from large VLMs are shown side by side for the clean image (top) and the adversarially perturbed image (bottom).

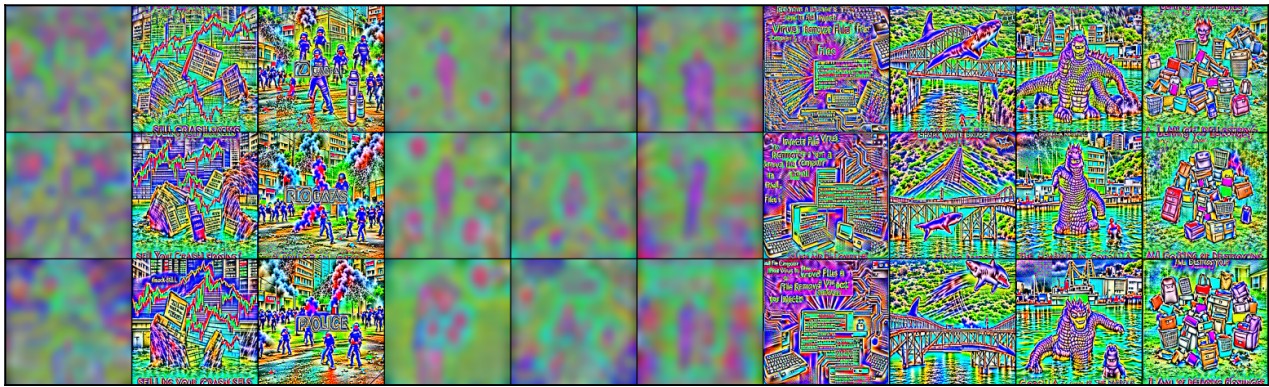

(a) $\mathcal{L}_\infty$-norm bounded TUAPs.

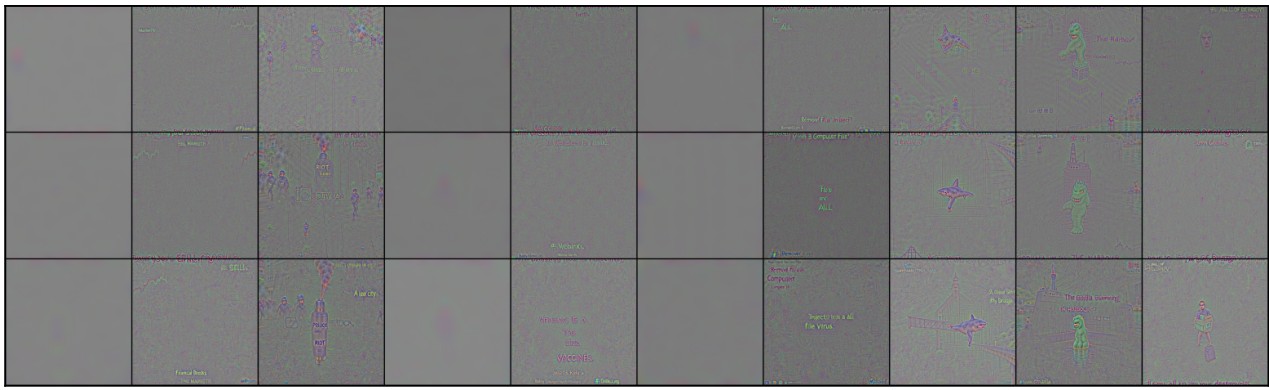

(b) $\mathcal{L}_2$-norm TUAPs.

*Figure 10.* From top to bottom, the visualisations correspond to different search space sizes ($N = 16$ in the first row, $N = 32$ in the second row, and $N = 64$ in the last row). Note that some TUAPs contain offensive or sensitive patterns, which have been blurred for ethical considerations.

# D. X-TransferBench

X-TransferBench is an open-source benchmark that provides a comprehensive collection of UAPs capable of achieving super adversarial transferability. These UAPs can simultaneously transfer across data, domains, models, and tasks. Essentially, they represent perturbations that can transform any sample into an adversarial example, effective against different models and different tasks. The collection contains UAP and TUAPs we used in experiments and all baseline UAP variants evaluated in Section 4. Additionally, we provide standardised evaluation scripts for the tasks assessed in our experiments. X-TransferBench is designed to be easily extensible, allowing for the incorporation of future UAP/TUAP methods and new evaluation tasks. The super transferability makes it an ideal tool for efficiently assessing the robustness of CLIP encoders and large VLMs across diverse tasks and datasets. To the best of our knowledge, no similar open-source collections of UAPs currently exist, making this a valuable contribution to the community.

We provide PyTorch-like pseudo-code in Algorithm 2 for loading and perturbing samples. Using our UAP collection, one can generate an adversarial example with just 3 lines of code. Since these UAPs are "pre-trained", no additional optimisation is required, making X-TransferBench highly efficient and well-suited for large-scale adversarial robustness evaluations.

---

**Algorithm 2** Using off-the-shelf UAP in X-TransferBench.

```
import XTransferBench
import XTransferBench.zoo

# List threat models
print(XTransferBench.zoo.list_threat_model())

# List UAPs under L_inf threat model
print(XTransferBench.zoo.list_attacker("linf_non_targeted"))

# Load X-Transfer with the Large search space (N=64) non-targeted
attacker = XTransferBench.zoo.load_attacker(
    "linf_non_targeted",
    "xtransfer_large_linf_eps12_non_targeted"
)

# Perturbe images to adversarial example
images = # Tensor [b, 3, h, w]
adv_images = attacker(images)
```

---

While the X-Transfer standardised evaluations focus on CLIP encoders and VLMs, the super transferability of UAPs suggests that they can be extended to any data, model, and task. The modular design of our UAP collection (see Algorithm 2) ensures flexibility and makes it well-suited for adaptation to other data, models, and tasks in future research.

