# OpenReview forum: "X-Transfer Attacks: Towards Super Transferable Adversarial Attacks on CLIP"
_ICML.cc/2025/Conference — ICML 2025 poster_

### Official Review · Reviewer_HNj3 · 2025-03-10

**Overall Recommendation:** 3

**Summary:**

With wide-spread deployment of CLIP image- and text-encoding for use in downstream tasks, CLIP has been revealed as a useful attack surface for subverting inference in the downstream model. A natural question is how to craft an effective perturbation once by way of ensemble methods which then subverts many models given any data point (i.e., universal adversarial perturbation (UAP)). The authors extend this idea to the realm of subverting many models, tasks, datasets, and domains, and describe such transferability as so-called super transferability taking form as super universal adversarial perturbations (SUAP). Previously, UAP methods require a fixed set of models, heuristics, and constraints to optimize a UAP with desired effect. The authors argue that this becomes intractable when tackling the problem of super-transferability, and instead propose to use heuristics agnostic to the surrogate model. In practice, the authors use the heuristic which simply minimizes (or maximizes) the similarity in CLIP embedding space of untargeted (or targeted) evasion attack. This heuristic works since all downstream tasks will exploit the contrastive nature of CLIP embeddings in some way. To scale up to many surrogate models, the authors finally propose to adopt upper confidence bounds (UCB) from the well-studied multi-armed bandits framework to avoid optimizing with respect to all surrogate models at once, instead letting the UCB-based algorithm (X-Transfer) guide the sampling of surrogates. Specifically, the authors cast the UAP optimization over surrogate models as a non-stationary multi-armed bandits problem, since the reward distribution for different surrogate models may shift as optimization progresses.

## Post-rebuttal update

Thanks to the authors for their response. They will incorporate some of the changes mentioned in the original review, and have provided some interpretations of the submission's pain points. I will keep my original score.

**Claims And Evidence:**

- The authors suggest that by efficient selection of surrogate models during ensemble optimization, the adversary may learn a super UAP which transfers to many tasks, models, datasets, and modalities. The main claim is checked through empirical studies starting in Section 4 for zero-shot classification (Table 1) and captioning/VQA (Table 2). In these tables it can be observed that X-Transfer performance improves as a larger set of surrogates is selected (e.g., going from Vanilla to Large offers reliable improvement of ASR). The effect of UCB selection is ablated in Figure 2a showing that (1) more surrogates leads to higher ASR, and (2) selection guided by UCB offers a compute reduction while matching the results of higher $k$.
- The evaluation suggests that UCB-based X-Transfer may reduce the computational cost of SUAP. However, it is also shown that both random sampling and $\epsilon$-greedy sampling strategies have comparable performance, hence it might be difficult to motivate the extra complexity of using UCB over simple random sampling. The main takeaway for this result is that sub-sampling of surrogate models is important to achieve SUAP. As a further takeaway, it could be argued that since UCB does not have a large gap compared to random sampling, the reward distribution of the surrogate models may be more trivial (i.e., stationary) than expected. It could be that in the case of dynamic surrogates (i.e., updated in real time against the adversary) the reward distribution becomes more suitable for UCB.

**Essential References Not Discussed:**

N/A

**Experimental Designs Or Analyses:**

- The experiments use a variety of established baselines such as C-PGC, ETU, and AdvCLIP for CLIP-specific attacks, and also attacks meant as general purpose UAP, such as GD-UAP, TRM-UAP, and Meta-UAP. An ablation is performed which is the same attack setup of X-Transfer but without the UCB-guided surrogate selection. These baselines are reasonable due to being relatively recent but also relevant to the goal (attacking CLIP models).
- Cross-data experiments operate on standard benchmark datasets which vary in scale, fidelity, and purpose. Cross-model experiments use a mix of both ResNet- and transformer-backed CLIP models which isolates causes due to model architecture. Other more recent methods are checked, such as MetaCLIP, meanwhile some VLMs are selected to study transferability on multi-modal LLM inference. In this latter case the authors do not launch X-Transfer on the respective VLM finetune of CLIP, which seems reasonable.

**Methods And Evaluation Criteria:**

- The methods for evaluating each task (e.g., CIDEr for image captioning) generally match the metric used to score the task in the respective literature. Attack success rate is used for all tables which is the standard for adversarial ML studies.
- The benchmark datasets and associated models are reasonable since they are fairly recent approaches.

**Other Comments Or Suggestions:**

* Equation 1 - the text should connect $k$ and $j$ to the contrasting of image with texts and the reverse for clarity.
* Equation 4 - $t_{adv}$ should be clarified to be a sample of interest.
* L179-180 - encodes -> encoders

**Other Strengths And Weaknesses:**

- Since the ablation shows that UCB is marginally better than random sampling for the fixed CLIP models, it might be interesting to check the reward distribution of CLIP models which update dynamically against the adversary (hence changing the reward distribution over course of adversary's training loop). It could be that UCB becomes more useful on an adaptive defender threat model, where the CLIP model is updated on-the-fly during adversary's optimization loop (to mimic a defender's inner update loop).
- The writing is generally high quality and easy to follow for someone familiar with the prior work.
- The approach could be considered an iterative improvement over prior work since it mainly combines UCB with known objectives and optimization algorithm. I would consider the primary contribution to be the ablations and empirical results studying ensemble sampling behavior.

**Questions For Authors:**

N/A

**Relation To Broader Scientific Literature:**

- The authors plan to release a large collection of UAPs and TUAPs to the broader community, which may be useful for checking future defenses on a static target.
- The authors show that subsampling of the surrogate ensemble is a simple but effective way of improving UAPs. A variety of baseline methods and surrogate models are checked which will serve as useful reported baselines in future studies.

**Theoretical Claims:**

N/A

---

> ### Author Rebuttal · Authors · 2025-03-30
>
> We are grateful for your careful review of our paper. Please find our detailed responses to your questions below.
>
> ---
>
> **Q1:** Choice of sampling strategies and improvement over random sampling
>
> **A1:**
> We would like to clarify and emphasize the comparison of sampling strategies presented in Section 4.2. The results are reported as macro-averages across 9 encoders, each evaluated on 8 datasets, yielding a total of 72 evaluation points. We argue that the observed 2.3% performance gap is statistically significant, particularly in the context of large-scale benchmarking.
>
> Our analysis demonstrates that reward-aware sampling strategies (UCB and $\epsilon$-greedy) consistently outperform reward-agnostic random sampling. This empirical evidence strongly suggests that the reward metric itself - rather than the sampling strategy choice - is the primary driver of super-transferability improvements. While we acknowledge that alternative sampling approaches may be worth exploring, our core contributions remain: (1) the efficient surrogate scaling framework and (2) the proposed reward metric.
>
> To summarise:
> - Super-transferability stems fundamentally from surrogate scaling, as evidenced by experiments with varying search space sizes ($N$).
> - While sub-sampling improves computational efficiency (Fig 2a), it is not the primary driver of super transferability.
> - Reward-aware sampling consistently outperforms random sampling while maintaining efficiency.
>
> ---
>
> **Q2:** The dynamic reward distribution
>
> **A2:**
> We would like to clarify that our current approach already incorporates a dynamic reward distribution that evolves throughout the adversary’s training loop. As the perturbation improves over time, it influences the loss values used as the reward signal, leading to a naturally shifting reward landscape.
>
> We also agree with the reviewer that exploring an adaptive threat model, in which CLIP itself is updated during the adversarial process, presents an interesting direction for future research. We appreciate this insightful and constructive suggestion and will certainly pursue it in future work.
>
> ---
>
> **Q3:** Iterative improvement and contribution
>
> **A3:**
> While the adversarial objective we employ may appear simple, its design is both intentional and necessary, and its combination with UCB may seem like an incremental improvement. However, we emphasize that this formulation has led to insightful and novel findings regarding the vulnerabilities of CLIP and VLMs. The simplicity and generality of the adversarial objective are deliberate design choices, as they ensure compatibility across diverse CLIP encoders and are crucial for achieving super transferability, as demonstrated in Appendix C.4. This deliberate simplicity underscores the elegance and effectiveness of our approach.
>
> Beyond performance, our work reveals a new and exciting phenomenon: the untargeted UAPs generated by X-Transfer are semantically interpretable, yet they do not align with human perception. Prior work has suggested that semantically interpretable perturbations that align with human understanding typically arise only from targeted objectives, whereas untargeted perturbations appear patternless. In contrast, our findings show that untargeted UAPs can also produce visually coherent patterns that are semantically rich but misaligned with human interpretation, suggesting that these perturbations explore a space distinct from all previously observed perturbation behaviors.
>
> We believe that a simple and effective method, paired with a novel and well-supported finding, and validated through rigorous experimentation, constitutes a meaningful and impactful contribution to the field.
>
> ---
> **R4:** Others suggestions
>
> **A4:** We will incorporate these suggestions in the revision.

---

### Official Review · Reviewer_GY9n · 2025-03-13

**Overall Recommendation:** 2

**Summary:**

This paper reveals a universal adversarial vulnerability in CLIP models, where a single perturbation achieves super-transferability across datasets, domains, models, and tasks. The authors find that proxy encoder selection, rather than the dataset, is the key factor. They propose X-Transfer, a novel attack that surpasses prior UAP methods in transferability, establishing a new benchmark. Additionally, they introduce X-TransferBench, an open-source evaluation framework for CLIP and VLM robustness. The core innovation lies in an efficient proxy scaling strategy, leveraging UCB sampling to optimize encoder selection, enhance perturbation generalization, and achieve super-transferability.

**Claims And Evidence:**

Yes.

**Essential References Not Discussed:**

No.

**Experimental Designs Or Analyses:**

(1) Key hyperparameters, such as the proxy encoder search space size and the number of selected encoders per iteration, are empirically set without rigorous theoretical justification or systematic validation.
(2) The study relies solely on OpenCLIP, which may not fully capture the diversity of real-world CLIP deployments.

**Methods And Evaluation Criteria:**

Yes.

**Other Comments Or Suggestions:**

Developing defenses against X-Transfer is essential. It should explore robust training, architectural modifications, and specialized defenses while balancing security and model performance.

**Other Strengths And Weaknesses:**

(1) X-Transfer is empirically validated, but key theoretical explanations remain lacking. The dominance of ViT-based encoders in proxy selection is observed but not deeply analyzed, and the emergence of semantic patterns in UAPs is attributed to CLIP’s concept fusion ability without solid theoretical grounding.
(2) Despite extensive experiments, real-world deployment factors—such as data formats and dynamic model environments—are not fully considered.
(3) Key hyperparameters, including search space size and proxy encoder selection, are determined heuristically without rigorous analysis. Relying solely on empirical tuning limits the method’s generalizability and scalability.

**Questions For Authors:**

No.

**Relation To Broader Scientific Literature:**

(1) CLIP’s contrastive learning enables strong zero-shot generalization, driving research into its applications and vulnerabilities.
(2) While UAPs have been explored for CLIP, prior methods struggle with super-transferability across datasets, domains, models, and tasks. X-Transfer fills this gap, setting a new benchmark for UAP effectiveness in adversarial attacks on CLIP.

**Theoretical Claims:**

(1) The study primarily relies on OpenCLIP, which, while widely used, may not fully capture the diversity of CLIP models in real-world applications. Differences in architecture, training, and optimizations across implementations could impact X-Transfer’s effectiveness, limiting the generalizability of the findings.
(2) Although the evaluation covers various tasks and datasets, it lacks real-world complexity. Practical deployment factors—such as data formats, model updates, distribution shifts, and hardware constraints—may affect X-Transfer’s performance.

---

> ### Author Rebuttal · Authors · 2025-03-30
>
> We sincerely appreciate your time and careful review of our work. Below, we provide detailed responses to address each of your concerns.
>
> ---
>
> **Q1:** Using OpenCLIP lacks generalisability and real-world models.
>
> **A1:**
>
> (1) OpenCLIP is an open-source framework, protocol, and API for CLIP and its variants, not a model itself. It currently hosts the largest collection of open-source CLIP models, which we adopt in our experiments primarily for its convenient and unified interface. Many real-world CLIP models are already included in the OpenCLIP collection, such as MetaCLIP from Meta and SigLIP from Google, both of which are evaluated in our experiments.
>
> (2) To address your concern regarding real-world deployment, we conducted evaluations on GPT-4o and OpenAI o1. Using the same targeted adversarial example generated via TUAP (as shown in Figure 3), we prompted the models with *“Briefly describe the image.”* Below are the models’ responses.
>
> **GPT4-o**: *The image is a digital collage blending a volcanic eruption with cyber elements. Lava flows into the ocean under a blue-toned overlay of circuit patterns and glowing text related to computer viruses, such as “Remove Files” and “Virus Inject.” It visually represents the destructive power of cyber threats.*
>
> **OpenAI o1**: *It’s a composite image showing lava pouring into an ocean, overlaid with circuit-board patterns and words referencing computer viruses and file removal, evoking the idea of a powerful, destructive cyber threat.*
>
> We also conducted a qualitative evaluation by randomly sampling 1,000 images from the ImageNet test set and applying our TUAP to generate adversarial examples. We queried the model using the prompt “Briefly describe the image.” For this evaluation, we used OpenAI’s GPT-4o-mini model. Each request included both the textual prompt and the adversarial image (encoded in base64), and the maximum response length was set to 128 tokens. If the response contained any of the keywords in target text descriptions (e.g. *“remove”, “files”, “computer”, “inject”, “computer virus”*), we considered it a successful attack. The ASR based on this criterion is reported in the table below.
>
> | Base | Mid | Large |
> |---|---|---|
> | 53.6 | 65.8 | 70.0 |
>
> ---
>
> **Q2:** Ablation on hyperparameters
>
> **A2:**
> The search space size, denoted as $N$, is evaluated throughout the paper, and the number of selected encoders per iteration, $k$, is specifically analysed in Section 4.2.
>
> Results for different search space sizes, Vanilla ($N$ = 1), Base ($N$ = 16), Mid ($N$ = 32), and Large ($N$ = 64), are presented in Tables 1, 2, 13, 15, and 16. These results demonstrate that a larger search space consistently improves the ASR.
>
> The ablation study on $k$ is shown in Figure 2(a), and the efficiency analysis is provided in Table 7. The results indicate that while varying $k$ has minimal impact on ASR, it primarily affects computational efficiency. Our choice of $k$ is 25% of $N$ is motivated by the trade-off observed in Figure 2(a) and the efficiency analysis in Table 7.
>
> ---
>
> **Q3:** Theoretical explanations
>
> **A3:**
> We believe that super transferability and CLIP’s concept fusion ability are novel topics of growing importance. As such, there are currently no established theoretical frameworks to formally characterise these phenomena. Nevertheless, our empirical results are comprehensive, providing strong support for our claims regarding super transferability. The novel observation that UAPs on CLIP exhibit semantically interpretable patterns offers valuable insight into the underlying vulnerabilities of CLIP models. We plan to investigate theoretical explanations in our future work.
>
> ---
>
> **Q4:** Defences against X-Transfer
>
> **A4:**
> Please refer to Appendix C.8, where we have presented evaluations using adversarially trained CLIP encoders.

---

### Official Review · Reviewer_wapn · 2025-03-13

**Overall Recommendation:** 5

**Summary:**

This paper introduces X-Transfer, a novel adversarial attack method that generates universal adversarial perturbations (UAPs) with "super transferability" across data, domains, models, and tasks for CLIP-based vision-language models. The core innovation is an efficient surrogate scaling strategy that dynamically selects a subset of surrogate CLIP encoders from a large search space using a multi-armed bandit (MAB) framework with Upper Confidence Bound (UCB) sampling. Extensive experiments demonstrate that X-Transfer outperforms state-of-the-art UAP methods, achieving higher attack success rates (ASR) on zero-shot classification, image-text retrieval, image captioning, and VQA tasks. The authors also release X-TransferBench, a comprehensive benchmark of UAPs for evaluating super transferability.

**Claims And Evidence:**

- Claim 1: X-Transfer achieves "super transferability" (simultaneous cross-data/domain/model/task transferability).
  - Evidence: Supported by experiments across 12 datasets, 9 CLIP encoders, and 4 VLMs (Table 1-2). However, cross-task transferability (e.g., attacking VLMs trained with autoregressive objectives) lacks mechanistic explanation.

- Claim 2: The dynamic surrogate selection strategy reduces computational costs while improving transferability.
  - Evidence: Figure 2(a) shows X-Transfer with $k=1$ matches standard scaling ($k=N$), but theoretical guarantees for UCB-based selection (e.g., regret bounds) are missing.

- Claim 3: X-TransferBench provides a practical resource for adversarial robustness evaluation.
  - Evidence: The benchmark is described in Appendix D, but no user studies or community adoption examples are provided.

**Essential References Not Discussed:**

- Multi-modal adversarial attacks: Concurrent work on GPT-4V/DALL-E 3 adversarial attacks (e.g., "ImgTrojan: Jailbreaking Vision-Language Models with ONE Image" is not cited.
- Theoretical analysis of UAPs: The paper does not discuss recent theoretical frameworks for UAP transferability (e.g., "On the Universal Adversarial Perturbations for Efficient Data-Free Robustness Evaluation" (ACL 2023)).

**Experimental Designs Or Analyses:**

- Strengths:
  - Broad evaluation across CLIP variants (ViT, ResNet, SigLIP) and VLMs (LLaVA, MiniGPT-4).
  - Ablation studies on surrogate scaling (Figure 2(a)) and perturbation types ($L_2$, patch).

- Weaknesses:
  - "Following Fang et al. (2024b); Zhang et al. (2024), we employ L∞-norm bounded perturbations with ϵ = 12/255, and the step size η of 0.5/255." I did find the description of ϵ = 12/255 in the two papers, but I did not find the description of step size η of 0.5/255. Please provide more detailed sources for the setting of step size.
  - The search space (Tables 8-10) includes only CLIP variants, excluding non-CLIP vision-language models (e.g., ALIGN, Florence).
  - Results on adversarial training (Appendix C.8) are superficial; no defense strategies (e.g., randomized smoothing) are discussed.

**Methods And Evaluation Criteria:**

- Methods:
  - The MAB-based dynamic selection is novel and addresses scalability limitations of fixed ensembles. However, the choice of UCB over other bandit algorithms (e.g., Thompson sampling) is not justified.
  - The adversarial objective (Eq. 3-5) is generic but lacks architectural/task-specific adaptations (e.g., for VLM autoregressive losses).

- Evaluation Criteria:
  - ASR is task-specific (e.g., accuracy drop for classification, CIDEr for captioning), which is reasonable. However, targeted attacks (TUAPs) are only evaluated on a limited set of 10 manually designed text prompts (Appendix C.7), raising concerns about generalizability.

**Other Comments Or Suggestions:**

Please provide more examples similar to Figure 3.

**Other Strengths And Weaknesses:**

- Strengths:
  - Originality: First to formalize "super transferability" and propose a scalable solution.
  - Practical Impact: X-TransferBench fills a gap in standardized UAP evaluation.

- Weaknesses:
  - Clarity: Equation symbols (e.g., $\mathbb{D}$ vs. $\mathbb{D}$) are inconsistently defined.
  - Significance: While results on existing CLIP/VLMs are strong, applicability to newer models is unclear.

**Questions For Authors:**

1. Theoretical Justification: Can you provide regret bounds or convergence guarantees for the UCB-based surrogate selection strategy? *A theoretical analysis would strengthen the method’s credibility.*
2. Search Space Generalization: How does X-Transfer perform if the search space contains only ViT or ResNet architectures? *This would test the robustness of dynamic selection to model homogeneity.*
3. Defense Evaluation: Have you evaluated X-Transfer against state-of-the-art defenses (e.g., randomized smoothing or feature denoising)? *This would clarify the practical threat model.*

**Relation To Broader Scientific Literature:**

- Extends prior work on UAPs (Moosavi-Dezfooli et al., 2017) and CLIP adversarial attacks (Zhou et al., 2023; Zhang et al., 2024) by addressing multi-dimensional transferability.
- Connects to bandit algorithms (Auer, 2002) but does not leverage recent advances in contextual bandits for non-stationary environments.

**Theoretical Claims:**

- The paper lacks theoretical analysis. For example:
  - No proof of convergence for the UCB-based selection strategy.
  - No formal analysis of why ViT-based surrogates dominate selection (Figure 2(b)) or how embedding space geometry enables cross-task transfer.

---

> ### Author Rebuttal · Authors · 2025-03-30
>
> We sincerely appreciate your thorough review and insightful comments. Please find our responses to your questions below.
>
> **Q1:** Mechanism behind the cross-task transferability on VLM
>
> **A1:** X-Transfer exploits a common weakness in CLIP image encoders, even when they are trained on different datasets and model architectures. Our adversarial objective is intentionally generic as it induces **meaningless embeddings**. Since many VLMs adopt CLIP or its variants as their image encoders, the UAPs generated by X-Transfer can therefore induce similar **nonsensical embeddings** in these models. This highlights a shared vulnerability across a wide range of CLIP encoders.
>
> **Q2:** Theoretical analysis
>
> **A2:** The focus of this work is super transferability, not MAB or UCB regret analysis.
>
> We believe that super transferability is a novel topic, and as such, there is currently no established theoretical framework. However, our empirical results are solid and comprehensive, providing strong support for our claims on super transferability. The geometry-based theoretical analysis of transferability between classifiers (Tramèr et al., 2017), as well as the ACL23 work mentioned by the reviewer, could serve as valuable starting points for future theoretical investigations into super transferability.
>
> **Q3:** X-TransferBench and supplementary material
>
> **A3:** The sample code is included in the accompanying ZIP file. We will adopt the MIT License in the published version. Please refer to Figure 2(a) for the correlation with ASR.
>
> **Q4:** Choice sampling method and generic adversarial objective
>
> **A4:** Our main contribution lies in the MAB-based dynamic selection framework and the design of a reward metric. The sampling method itself is not the primary factor in achieving super transferability. It is the reward metric that plays a critical role (see Section 4.2). In Section 3.3, we stated that while UCB is our default choice due to its simplicity, we also emphasised that other sampling strategies (including Thompson sampling) are plausible and fully compatible with our framework.
>
> The generic objective (Eq. 3-5) is essential to ensure compatibility across different architectures, embedding sizes, and pre-training objectives of surrogate models. While the objectives may appear simple, they are crucial for achieving super transferability, see Appendix C.4 and Table 11.
>
> **Q5:** New experiments (TUAP, step size, other and newer models, randomised smoothing, and search space)
>
> **A5:**
>
> (1) We tested 5 additional TUAPs (average ASR 77.6%) with targets sampled from the AdvBench and compared their performance with the 10 TUAPs (average ASR 75.8%). The results are consistent.
>
> (2) We follow existing works and set the $\epsilon$ to 12/255. The step size is a typo. It is our choice of hyperparameters, not baselines. This does not affect our analyses since $\epsilon$ is the factor in ensuring the fair comparison for $L_\infty$ perturbations.
>
> (3) Please find the results below for ALIGN, Florence and newer models. Note MetaCLIP-v1.2 ViT-H/14 was released in Dec 2024, and SigLIP-v2 ViT-B-16 was released in Feb 2025.
>
> | Model | Task | ETU | C-GPC | Base | Mid | Large |
> |:---:|:---:|:---:|:---:|:---:|:---:|:---:|
> | ALIGN | ZS | 57.1 | 52.9 | 68.7 | 65.7 | **70.6** |
> |  | IR | 45.3 | 38.7 | 56.7 | 59.8 | **63.8** |
> |  | TR | 52.9 | 49.6 | 66.5 | 66.5 | **71.0** |
> | MetaCLIP-v1.2 | ZS | 36.0 | 33.9 | 61.0 | 64.1 | **70.5** |
> |  | IR | 15.2 | 20.3 | 40.5 | 43.4 | **48.4** |
> |  | TR | 27.7 | 29.6 | 48.8 | 55.0 | **61.9** |
> | SigLIP-v2 | ZS | 55.3 | 49.0 | 66.7 | 69.3 | **72.1** |
> |  | IR | 27.8 | 34.2 | 51.4 | 56.2 | **59.1** |
> |  | TR | 49.4 | 46.3 | 59.7 | 65.3 | **67.4** |
> | Florence-v2 | COCO-IC | 24.4 | 15.2 | 29.4 | 29.7 | **33.0** |
> |  | Flicker-30k-IC | 25.4 | 20.5 | 28.5 | 28.5 | **31.2** |
>
> While we cannot guarantee complete effectiveness against newer models, we believe our work offers novel insights into CLIP/VLM vulnerabilities and makes valuable contributions to the community. These findings provide important foundations for developing safer, more robust models in the future.
>
> (4) We tested our UAP against the smoothed ImageNet classifier provided by Cohen et al., 2019. As expected, the classifier demonstrates robustness to $L_\infty$ and $L_2$ perturbations, consistent with its certified guarantees. However, it is not robust to our patch-based UAP, as these perturbations lie outside the certified radius. This finding highlights an important gap and, we believe, serves as a strong motivation for future works.
>
> (5) We evaluated the generalisation capability of our method by testing with a ViT-only base search space (average ASR of 69.3%) and comparing it to the original mixed-architecture base search space (average ASR of 69.2%). The results show no significant difference.
>
> **Q6:** Others
>
> **A6:** We will fix the typo, add more examples of Fig 3, and add references to related works mentioned by the reviewer.

---

### Official Review · Reviewer_UCtW · 2025-03-13

**Overall Recommendation:** 5

**Summary:**

This paper introduces an algorithm to find universal adversarial perturbations for CLIP-like image encoders. The vulnerability works across domains, tasks, and samples. The main algorithm follows standard methods for finding adversarial perturbations: finding a perturbation whose \( L_{\infty} \) norm is smaller than a given \( \epsilon \) while also optimizing the attack objective so that the CLIP encoding becomes similar to a text encoding for a certain malicious input text. To make this perturbation universal, the authors use multiple encoders.

The main novelty of the attack lies in the efficient use of a large zoo of \( N \) CLIP-like image encoders for finding a universal perturbation. To this end, the authors use a reward-based strategy to select \( k \ll N \) encoders at each optimization step of the attack. The experiments are conducted on several tasks, including zero-shot classification, image-text retrieval, image captioning, and VQA, demonstrating the versatility of the attack.

The most interesting part of the paper is the finding that the vanilla version, without an ensemble of models (often a requirement for finding a universal perturbation), works quite well for classification and image-text retrieval tasks and actually outperforms several existing methods by large margins.

Another interesting insight from the paper is the appearance of text-like artifacts on perturbed images.

Overall, this is an excellent paper (the kind of paper I would love to write). It is easy to understand, makes claims backed by ample empirical evidence, demonstrates the power of simpler methods, and has an excellent evaluation setup and experiment/analysis.

**Claims And Evidence:**

The claims in this paper are clear and supported by ample empirical evidence. The main claim is the introduction of an efficient "super" UAP that works across tasks, models, and domains. This is backed by comprehensive experiments.

**Essential References Not Discussed:**

All essential references are discussed.

**Experimental Designs Or Analyses:**

I did not evaluate the experimental design by running any code. However, I compared the design with existing UAP methods and found it to be sound. Moreover, the results make sense, and the appearance of buildings and text on the input images indicates the validity of the experiments.

**Methods And Evaluation Criteria:**

The method and evaluation criteria presented in the paper align well with the problem of finding UAPs.

**Other Comments Or Suggestions:**

No.

**Other Strengths And Weaknesses:**

The paper is clear and presents a novel insight into the vulnerability of VLMs.

**Questions For Authors:**

The finding that the "vanilla version without an ensemble works well" has important implications. It suggests that a single-model attack can be surprisingly effective, challenging the common assumption that an ensemble is necessary for universal perturbations. Does this raise questions about whether previous attacks on CLIP were executed optimally or if they relied on unnecessarily complex setups? It may also indicate that CLIP-like models share inherent vulnerabilities that can be exploited more easily than previously thought.

The appearance of text on perturbed images is an interesting phenomenon. CLIP has been shown to be deceived by directly imposing text on input images, so this could suggest that the perturbations exploit similar weaknesses in the model’s reliance on textual features within visual inputs.

**Relation To Broader Scientific Literature:**

The main contribution for me is its finding that non-ensemble-based UAP with standard methods work quite well. This finding is

**Theoretical Claims:**

There are no theoretical claims in this paper.

---

> ### Author Rebuttal · Authors · 2025-03-30
>
> We sincerely appreciate your review, valuable feedback, and kind recognition of our work. Below are our responses to your questions.
>
> ---
>
> **Q1:** The common assumption that an ensemble is necessary.
>
> **A1:** We agree that it is indeed surprising that the vanilla version of X-Transfer without an ensemble, performs comparably to several strong baselines. However, we believe that the ensemble mechanism is crucial for achieving super transferability, as it allows the method to exploit shared vulnerabilities across diverse surrogate models.
>
> Regarding the complex setups used in prior work, we believe that they may be well-justified for the specific task those studies focused on. However, for the broader goal of super transferability, including cross-task scenarios, we find that a simple and generic adversarial objective is not only sufficient but also necessary. Notably, our experiments (Appendix C.4) demonstrate that surrogate scaling does not improve transferability when applied to the baseline (ETU), further reinforcing the motivation behind our design choices.
>
> ---
>
> **Q2:** Appearance of text features in perturbation
>
> **A2:** We would like to clarify that we did not use any explicit objective to introduce textual features or characters into the perturbation. Rather, these textual-like patterns **emerged naturally** as a byproduct of the optimisation process. The goal of X-Transfer is to exploit common vulnerabilities in CLIP encoders, and we believe our findings indeed demonstrate that these textual features reflect a shared vulnerability across CLIP variants. Understanding why such patterns emerge and how they interact with the multimodal representations of CLIP is, in our view, an interesting direction for future research.

---

### Decision · Program_Chairs · 2025-05-01

**Decision:**

Accept (poster)

**Comment:**

This work proposes a novel attack to generate universal adversarial perturbations against various CLIP encoders and downstream VLMs across different samples, tasks, and domains. The main novelty stems from an efficient surrogate scaling strategy that dynamically selects suitable surrogates. The experiments validate the effectiveness on different datasets and domains, outperforming sota baselines. Such a task and the results are meaningful and important in the community since CLIP has been widely used in diverse models and scenarios. Another interesting finding was noticed by the Reviewer UCtW: the vanilla version, without an ensemble of models (often a requirement for finding a universal perturbation), works quite well for classification and image-text retrieval tasks and actually outperforms several existing methods by large margins. Two reviewers rate the paper "strong accept" and agree that this work is excellent, with interesting findings, originality, and practical impacts, while one of the reviewers raises concerns about the lack of theoretical analysis. Another reviewer agrees that this work is well written and the primary contribution is the ablations and empirical results studying ensemble sampling behavior, tending to accept this work. One reviewer thinks the proposed method is a highly transferable universal adversarial attack on CLIP models, demonstrating its super-transferability. However, the reviewer highlights concerns about limited generalizability beyond OpenCLIP, heuristic-driven hyperparameters, and a lack of theoretical grounding, recommending further theoretical analysis and real-world validation. After reviewing the paper and reviewers' comments, I believe the first two concerns have been addressed: 1. OpenCLIP contains a wide range of CLIP models, and some of the models have been used in real applications. 2. The abolition studies have validated the influence of hyperparameters. Based on the above analysis and findings, I recommend accepting this work.